# HTLV-1 Tax plugs and freezes UPF1 helicase leading to nonsense-mediated mRNA decay inhibition

Francesca Fiorini[1,2,3], Jean-Philippe Robin[2], Joanne Kanaan[3], Malgorzata Borowiak[2,4], Vincent Croquette[5], Hervé Le Hir[3], Pierre Jalinot[2] & Vincent Mocquet[2]

Up-Frameshift Suppressor 1 Homolog (UPF1) is a key factor for nonsense-mediated mRNA decay (NMD), a cellular process that can actively degrade mRNAs. Here, we study NMD inhibition during infection by human T-cell lymphotropic virus type I (HTLV-1) and characterise the influence of the retroviral Tax factor on UPF1 activity. Tax interacts with the central helicase core domain of UPF1 and might plug the RNA channel of UPF1, reducing its affinity for nucleic acids. Furthermore, using a single-molecule approach, we show that the sequential interaction of Tax with a RNA-bound UPF1 freezes UPF1: this latter is less sensitive to the presence of ATP and shows translocation defects, highlighting the importance of this feature for NMD. These mechanistic insights reveal how HTLV-1 hijacks the central component of NMD to ensure expression of its own genome.

[1] Molecular Microbiology and Structural Biochemistry, MMSB-IBCP UMR5086 CNRS, Univ Lyon1, 7 passage du Vercors, 69367 Lyon Cedex 7, France. [2] Laboratory of Biology and Modelling of the Cell, ENS de Lyon, Univ Claude Bernard Lyon 1, CNRS UMR 5239, INSERM U1210, 46 allée d'Italie, 69364 Lyon, France. [3] Institut de Biologie de l'Ecole Normale Supérieure, CNRS UMR8197, Inserm, Ecole Normale Supérieure, PSL Research University, 46 rue d'Ulm, 75005 Paris, France. [4] Department of Chemistry and Pharmacy and Centre for Integrated Protein Science, Ludwig-Maximilians-University Munich, 5-13 Butenandtstrasse, 81377 Munich, Germany. [5] Laboratoire de Physique Statistique, École Normale Supérieure, PSL Research University, Univ Paris Diderot Sorbonne Paris-Cité, Sorbonne Univ UPMC Univ Paris 06, CNRS, 24 rue Lhomond, 75005 Paris, France. Francesca Fiorini and Jean-Philippe Robin contributed equally to this work. Correspondence and requests for materials should be addressed to P.J. (email: pjalinot@ens-lyon.fr) or to V.M. (email: vincent.mocquet@ens-lyon.fr)

Up-Frameshift Suppressor 1 Homolog (UPF1) is a DNA/RNA helicase at the crossroads of many critical cellular pathways for RNA and DNA maintenance, as well as for post-transcriptional regulation of gene expression. UPF1 is the central factor in nonsense-mediated mRNA decay (NMD) and is also directly involved in telomere homeostasis, DNA replication, histone mRNA degradation and staufen-mediated mRNA decay[1–4].

Our understanding of UPF1 action and regulation comes from the progressive elucidation of the molecular mechanisms involved in NMD. Despite the identification of several physiological and aberrant substrates, the mRNA features that trigger NMD are still elusive (reviewed in ref. [5]). A common working model for mammalian cells states that NMD is a process of targeting and degrading mRNA, depending on the composition of the RNP around the translation-terminating ribosome. Instead of contacting PABP1 for proper translation termination, the translation termination factors eRF1-3 associate with UPF1, which binds to RNA nonspecifically and accumulates downstream of the first stop codon to encounter the translating ribosome[6–11]. Hence, NMD initiates the decay of mRNA with a premature termination codon (PTC; induced by mutation, alternative splicing or frameshift); NMD also regulates the stability of non-mutated RNA, depending on the size and organisation of their 3′ untranslated region (3′UTR)[12–19]. Recently, NMD has been shown to have an important role in the pervasive and cryptic decay of transcripts in yeast, appearing as the major factor to remove spurious transcripts that have escaped degradation in the nucleus.

Stabilised at the stop codon, UPF1 then successfully activates its essential ATPase activity after contact with UPF2 and most likely other molecular partners[20–23]. Before decay occurs, UPF1 undergoes multiple SMG1-mediated phosphorylation events stimulated by UPF2 and UPF3[20,24,25]. These events lead to efficient recruitment of the endonuclease SMG6, as well as the adaptors SMG5, SMG7 and PNRC2, which are connected to general decapping, deadenylation and exonucleolytic activities, including XRN1[26–29]. It has been suggested that RNP remodelling during these late phases of the process most likely requires UPF1 translocation[30]. In mammals, NMD is enhanced by the 3′ presence of an exon junction complex (EJC) that facilitates UPF1 activation by its UPF2 and UPF3 components[20,23,31].

UPF1 is a modular enzyme that contains a conserved helicase core (helicase domain—HD) formed by two RecA lobes that are able to progressively unwind double-stranded nucleic acids through an inchworm mechanism of translocation[20,32,33]. The HD of the human protein is surrounded by two terminal domains: the N-terminal CH domain, which is enriched with cysteine and histidine residues, and the C-terminal SQ domain, which is enriched with clusters of serine–glutamine residues. Both domains tightly repress the ATPase and helicase activities of UPF1, raising the possibility of enzymatic activation during NMD[20–22]. Indeed, the RNA-dependent ATPase activity is required for NMD target selectivity and is essential for 3′mRNP remodelling and decay completion[30,34,35]. Structural and functional information describes how the CH domain, localised above the RecA2 domains, induces clamping of the enzyme on its RNA substrate by inhibiting its helicase and ATPase activities[20,21]. Binding of the UPF2 cofactor displaces the CH domain from its original position, thereby releasing UPF1 enzymatic activity[21,36]. An inhibitory function is also exerted by the SQ domain, regardless of its phosphorylation state[22]. Once triggered, the exact activity exerted by UPF1 on the RNA and the importance of its translocation during NMD are actually unknown.

There is now much evidence demonstrating the antiviral functions of NMD and the viral means to escape this cellular control (reviewed in refs. [37,38]). Plant and animal positive-strand RNA viruses, as well as retroviruses, use different mechanisms to fully express their compact genome that generate mRNA with NMD-inducing features[39–43]. A well-characterised example is exhibited by the RNA stability elements (RSEs) downstream of the translation termination codon of avian retrovirus Rous sarcoma virus, which stabilises its *gag* RNA[44–46]. These RSEs recruit polypyrimidine tract binding protein 1 (PTBP1), thereby reducing UPF1 binding to the 3′UTR[47]. During human immunodeficiency virus infection, NMD inhibition might occur by a tethering mechanism in which UPF1 is hijacked to promote the nucleocytoplasmic export of vRNAs that enhance their stability[42,48,49]. Interestingly, in Moloney murine leukaemia virus, reverse transcriptase binds to eRF1 at the translation-terminating ribosome to prevent the binding of eRF3 and UPF1, thereby promoting read-through of the stop codon and preventing NMD[50].

In HTLV-1, the viral proteins Tax and Rex exhibit a NMD inhibitory effect[5,6], and we correlated Tax-mediated inhibition to a direct interaction with UPF1[41]. In the present study, we use a combination of ex vivo and in vitro bulk and single-molecule assays to demonstrate how Tax directly affects UPF1 function. We describe a two-level regulation of UPF1 function demonstrating a Tax effect (1) prior to the binding of UPF1 to RNA and (2) on an actively unwinding enzyme. We show a regulatory host/pathogen mechanism, in which an exogenous factor is able to affect UPF1 translocation, highlighting the importance of this mechanical feature for the NMD process.

## Results

**HTLV-1 Tax inhibits NMD.** In a previous report, we showed that, during HTLV-1 infection, viral RNAs are sensitive to NMD and that retroviral Tax protein can inhibit this important cellular mRNA surveillance pathway[41]. Here we analysed the stability of reporter and several endogenous NMD substrates in ex vivo experimental conditions simulating physiological HTLV-1 infection of the cell.

The decay assays were performed monitoring the mRNA levels by carrying out qRT-PCR analyses at the indicated time points after transcription inhibition induced by 100 µg.ml$^{-1}$ of 5,6-dichloro-1-D-ribofuranosylbenzimidazole (DRB). Using this assay the stability of mRNAs transcribed from transfected β-globin minigenes, WT or with a PTC in the second exon (hereafter called Gl-WT and Gl-PTC, respectively) showed a clear destabilization effect of the PTC (Fig. 1a) that was impaired by coexpression of Tax as previously reported (Supplementary Fig. 1a). In order to validate this PCR assay, the half-life of the PTC β-globin mRNA without and with Tax coexpression was also measured by performing a northern blot analysis. This approach led to a similar result that confirmed the Tax-stabilising effect (Supplementary Fig. 1b, c). This NMD impairment was similarly observed when cells were co-transfected with the plasmid pCMVHTLV1-WT that expresses the entire HTLV-1 genome instead of solely the Tax protein (Fig. 1b). The HTLV-1 genome possesses a region termed pX, located between the *env* gene and the 3′ LTR, which contains all the genes coding for regulatory viral factors including the Tax and Rex proteins[51]. While previous studies have shown that HTLV-1 Rex protein may also play a critical role in the suppression of host NMD activity[52], we engineered a molecular clone deleted of ~1 kb of the pX region coding the major part of the second exon of Tax and Rex (ΔpX molecular clone). In these conditions, Tax and Rex expression are abolished. Consistently with previous results, the mRNA decay assay and a P-bodies analysis showed that NMD efficiency was not affected in ΔpX-transfected cells (Fig. 1c and Supplementary Fig. 1d, e). To further assess the role of Tax, a rescue experiment

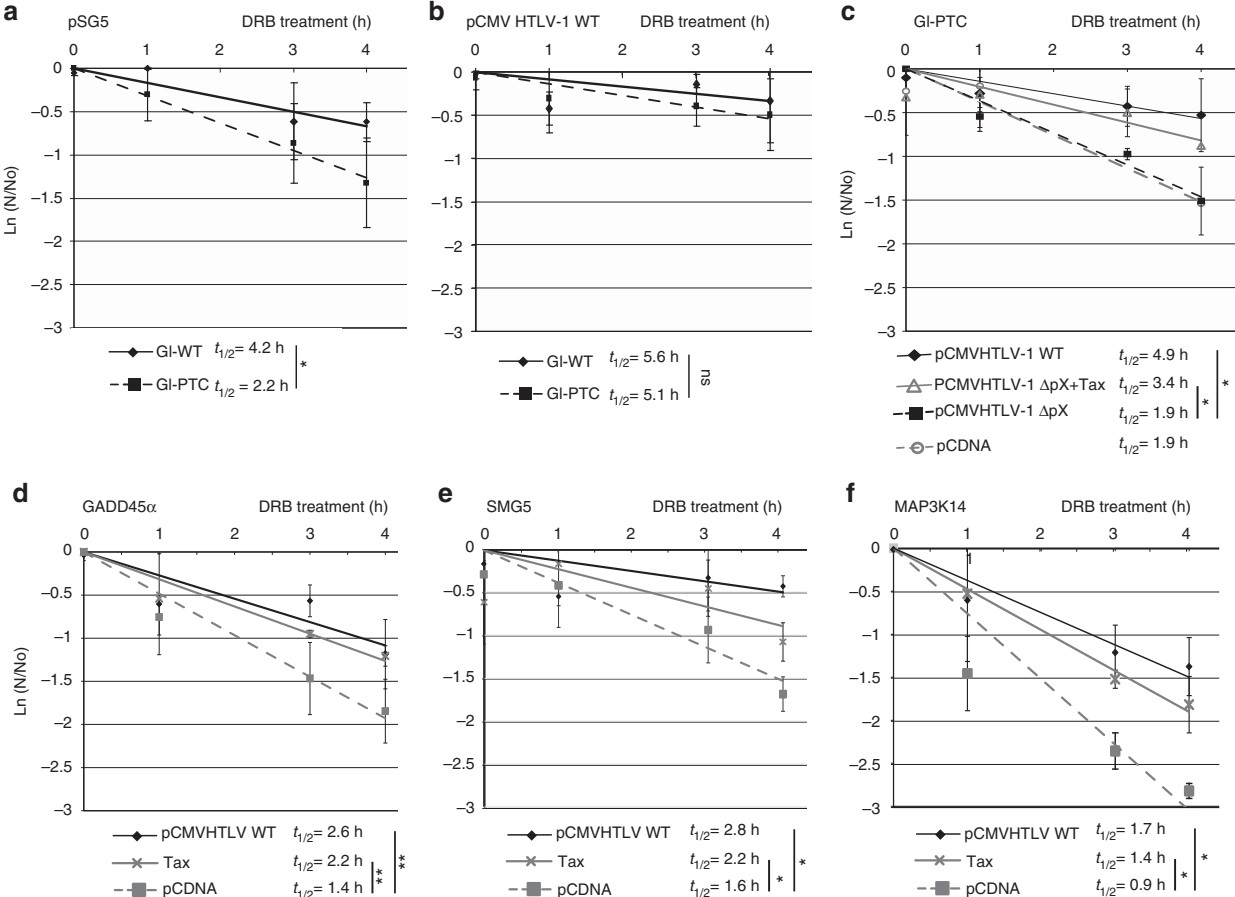

**Fig. 1** HTLV-1 Tax protects reporter and endogenous mRNAs from NMD in mammalian cells. **a** RNA decay assays were carried out in HeLa cells. The stability of GI-PTC and GI-WT mRNA was analysed after RNA quantification by qRT-PCR. mRNA half-lives ($t_{1/2} = \ln(2)/\lambda$ with $\lambda$ the time constant of the decay curves) are indicated in front of their respective conditions. **b** Same as **a** except that HeLa cells were co-transfected with HTLV-1 molecular clone. **c** RNA decay assays of GI-PTC mRNA in the presence of WT HTLV-1 molecular clone (continuous line, diamonds), truncated ΔpX HTLV-1 molecular clone complemented with Tax protein (continuous line, triangle), truncated ΔpX HTLV-1 molecular clone alone (dashed line, square) and empty vector (dashed line, circle). **d** RNA decay assays examining the stability of endogenous *GADD45α* mRNA in the presence of an empty vector (dashed line, square), Tax (continuous line, cross) or a WT HTLV-1 molecular clone (continuous line, diamond; **e**), **f** Same as **d** with *SMG5* and *MAP3K14* endogenous mRNA. The values represented in each graph correspond to the mean of at least three biological replicates, and the error bars correspond to the SD. Half-lives were calculated for each replicate, and *P* values were calculated by performing a Student's *t*-test (unpaired, two-tailed) ns: $P > 0.05$; *$P < 0.05$; **$P < 0.01$

was performed by coexpressing Tax concomitantly to pCMVHTLV-1 ΔpX transfection. The presence of Tax almost completely rescued stabilisation of the GI-PTC mRNA and thus restored NMD inhibition (Fig. 1c). In addition, we confirmed that this effect is not due to an inhibition of the translation since the protein neo-synthesis was not significantly modified under Tax expression (Supplementary Fig. 1f).

Then, under the same conditions, we analysed the stability of several NMD-prone endogenous mRNAs: GADD45α, SMG5 and MAP3K14 (Figs. 1d–f). GADD45α includes a 5′uORF and is one critical target of NMD in both *Drosophila* and mammalian cells[53,54]. SMG5 has been previously demonstrated to be NMD-sensitive due to its long 3′UTR, while the sensitivity of MAP3K14 is likely due to alternative splicing of exon11 in HeLa cells that leads to out-of-frame translation and occurrence of a PTC located more than 55nt- upstream of EJC (Genebank: CR749592.1 from clone DKFZp686J04131, cDNA sequencing consortium of the German Genome Project). The stability of these three endogenous NMD-prone mRNAs were increased by expression of Tax alone or transfection of the HTLV-1 molecular clone (Figs. 1d–f).

Taken together, these results confirm the direct role of Tax in NMD inhibition when the provirus is expressed and suggest a more general role of this factor in host gene expression.

**Tax directly binds UPF1-HD and inhibits its ATPase activity.**
In a previous work, we showed that Tax co-immunoprecipitated with several NMD factors, including the RNA helicase UPF1[41]. This result suggested that Tax-mediated inhibition of NMD could be related to its interaction with UPF1. To further test this hypothesis, we studied the Tax–UPF1 interaction using purified recombinant proteins and protein fragments (Fig. 2a). Concerning UPF1 recombinant proteins, the boundaries between the N-terminal (CH) domain, the central HD and the C-terminal (SQ) domain were defined according to previous structural studies[21,55]. We produced UPF1 full-length (UPF1-FL; amino acids (aa) 1−1118); a protein containing the CH domain of UPF1 (UPF1-CH; aa 115−294); UPF1 helicase core domain (UPF1-HD; aa 295−914); a protein fragment containing both the CH and HD domains (UPF1-CH-HD; aa 115−914); and a protein containing both the HD and the SQ domains (UPF1-HD-SQ; 295−1118 aa).

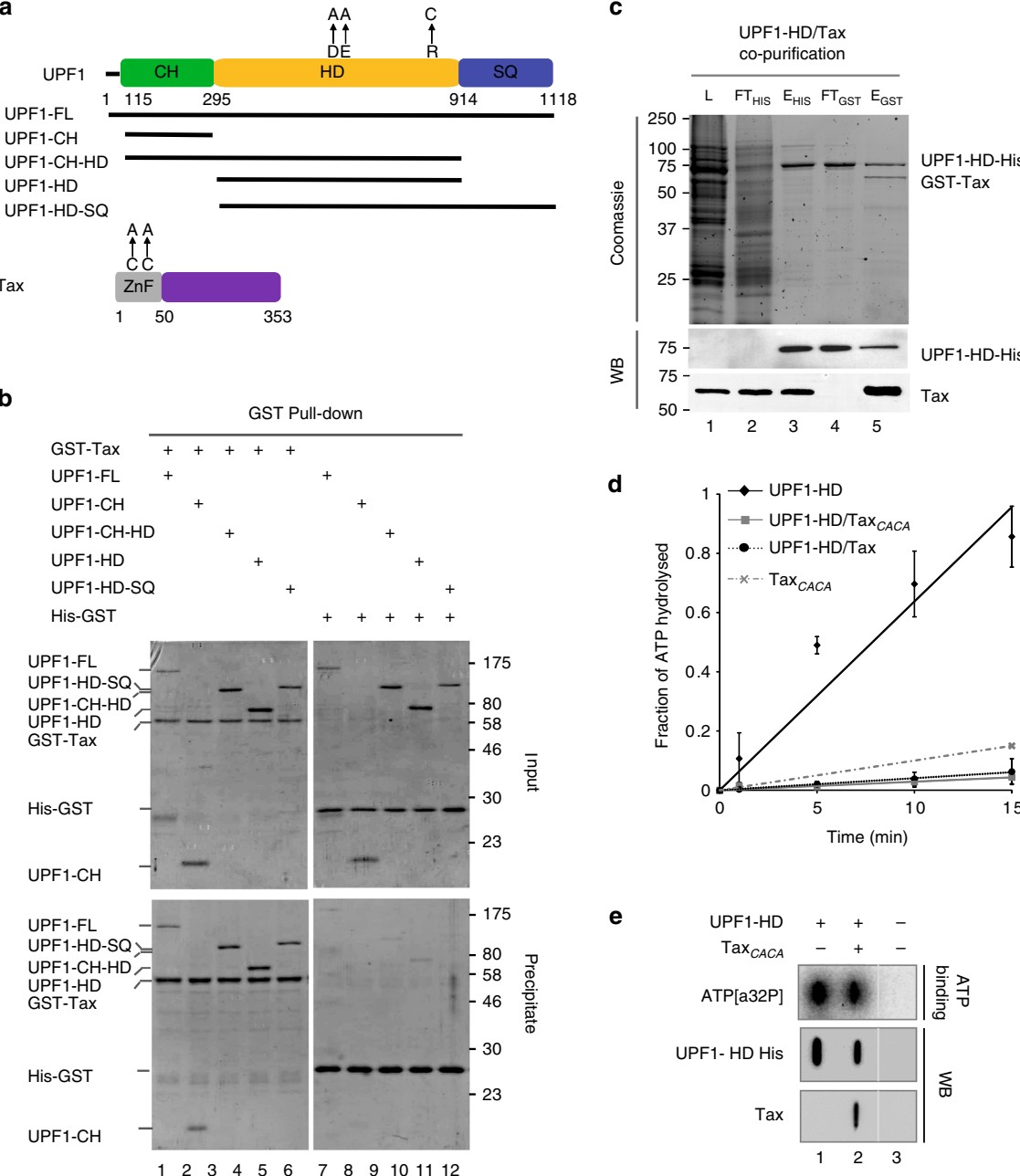

**Fig. 2** HTLV-1 Tax interacts directly with UPF1-HD and inhibits its ATPase activity. **a** Schematic diagram showing the UPF1 and Tax domains and sites of mutant derivatives used for this study. Structural domains are represented by rectangles and the protein truncations used by black lines. **b** Pulldown experiment using GST-Tax (lanes 1–6) or GST-His tag (lanes 7–12) as bait. After incubation, protein mixtures before (input 20% of total) or after precipitation (precipitate) were separated on a 10% SDS-PAGE gel and visualised by coomassie staining. **c** A SDS-PAGE gel illustrating the co-purification of human UPF1-HD-His and HTLV-1 GST-Tax proteins by two sequential affinity purifications (Nickel and Glutathione columns). Samples from the *Escherichia coli* lysate (lane 1: L), flow through (lanes 2 and 4: FTHIS and FTGST) and eluate (lanes 3 and 5: EHIS and EGST) fractions were loaded on the gel. The purity of the fractions was evaluated by coomassie staining (upper panel). The proteins were visualised by western blot analysis using anti-His and anti-Tax antibodies targeting UPF1 HD-His and GST-Tax, respectively (lower panels). **d** Graph showing the percentage of [α-$^{32}$P]ATP hydrolysed as a function of time by UPF1-HD (continuous line, diamond), UPF1-HD/Tax$_{CACA}$ (continuous line, square), UPF1-HD/Tax (dotted line, circle) and Tax$_{CACA}$ (dashed lane, cross) under conditions of steady-state ATPase turnover. Aliquots from the reaction mixture were quenched 0, 1, 5, 10 and 15 min before TLC chromatography (see Methods). The values are the mean of at least three biological replicates, and the error bars correspond to the SD. **e** UPF1-HD, UPF1-HD/Tax$_{CACA}$ complex and BSA proteins were spotted onto a nitrocellulose membrane and exposed to [α-$^{32}$P]ATP (top panel). The levels of the proteins used were also analysed by slot-blot on a different membrane. The membrane was incubated with anti-His and anti-Tax antibodies (middle and lower). Nonsignificant lanes were removed as indicated by the vertical white line. Uncropped scans related to Fig. 2 are available in Supplementary Fig. 6

Several point mutations, the position of which are indicated in Fig. 2a, were also produced for the purposes described thereafter. HTLV-1 Tax is a 353-amino-acid protein containing cysteine and histidine-rich regions at its N terminus (aa 22–53). UPF1 proteins were fused to a calmodulin-binding peptide at the N terminus and/or to a hexahistidine tag at the C terminus and were purified by successive affinity steps with nickel and calmodulin resin columns[22,56]. Analogously, N-terminal glutathione S-transferase (GST)-fused Tax was purified from a Glutathione Sepharose column as described elsewhere[57]. To assess the direct interaction of Tax with UPF1 and to map the eventual Tax-binding site in UPF1, we incubated the different UPF1 fragments with Tax and performed GST pulldown (Fig. 2b). After extensive washes with 0.3 M NaCl, input and eluted protein(s) were fractionated by sodium dodecyl sulphate-polyacrylamide gel electrophoresis (SDS-PAGE) and visualised by coomassie staining. UPF1-FL, UPF1-CH-HD, UPF1-HD and UPF1-HD-SQ were efficiently co-precipitated when GST-Tax was used as bait (lanes 1, 3–6) and compared to GST alone (lanes 7, 9–12). UPF1-CH interacted to a reduced extent compared with HD-containing proteins (lanes 2, 8 and Supplementary Fig. 2a). These data suggest that the Tax-binding site on UPF1 is most likely multipartite but clearly includes HD.

The effects of Tax on the NMD process together with its direct interaction with the UPF1 enzymatic core suggested possible inhibition of UPF1 activity. To prevent the nonspecific binding of $Tax_{WT}$ to RNA observed in RNA pull-down experiments, we engineered a double mutated version of Tax by changing C23 and C29 to alanine (mutant called $Tax_{CACA}$, Fig. 2a). We reasoned that disruption of the Zinc Finger (ZnF) domain in the Tax N-terminal region[58] should interfere with its nucleic acid binding. First, we controlled that this mutation is not altering Tax ability to inhibit NMD ex vivo (Supplementary Fig. 2c). In addition, overexpression of Tax protein in *E.coli* led to the formation of inclusion bodies. We speculated that the co-purification of UPF1-HD and Tax proteins might help to solubilise Tax. Indeed, using C-terminal hexahistidine-fused UPF1-HD and N-terminal GST-tagged Tax, we obtained the bimolecular complex after sequential Histidine and Glutathione affinity chromatography under stringent conditions (0.3 M NaCl; Fig. 2c, lane 5). The presence of both partners was confirmed by western blot analysis using anti-Histidine and anti-Tax antibodies (Fig. 2c, bottom panels). Analogously to UPF1-HD/$Tax_{CACA}$ complex, we were able to co-purify UPF1-HD with $Tax_{WT}$ as described before (Supplementary Fig. 2b).

The enzymatic activity of UPF1-HD was then measured by following the steady-state ATP hydrolysis rates of UPF1-HD, UPF1-HD/$Tax_{CACA}$ and UPF1-HD/Tax preformed complexes, as well as $Tax_{CACA}$ alone (Fig. 2d). The purified proteins and complexes (Supplementary Fig. 2b) were pre-incubated with RNA substrate (polyU), and the reaction was initiated by addition of ATP. The amount of ADP released was analysed by thin layer chromatography (TLC), and the ATPase efficiencies were deduced from the proportion of ATP and ADP over time (Fig. 2d). Under these conditions, all UPF1–HD/Tax complexes showed very weak ATPase activity compared with UPF1-HD. To assess the ability of UPF1-HD to bind ATP in the presence of Tax, we performed an ATP-binding assay. We incubated the proteins spotted on nitrocellulose membrane with [$\alpha$-$^{32}$P]ATP (Fig. 2e). The results showed that both UPF1-HD and UPF1-HD/$Tax_{CACA}$ complexes could bind ATP (lanes 1 and 2).

These results show a strong inhibitory effect of Tax on UPF1 ATPase activity without affecting ATP binding.

**Tax decreases UPF1-HD binding to RNA.** The ATPase activity of UPF1 and its affinity for the nucleic-acid substrate were linked to classify UPF1 as an RNA/DNA-dependent ATPase. Thus, we wondered whether the effect of Tax on decreasing UPF1 ATPase activity was due to a Tax-mediated RNA-binding defect. First, we analysed the RNA association of UPF1 ex vivo by performing RNA immunoprecipitation (RIP) experiments using HeLa cells transiently co-transfected with the Gl-PTC and HTLV-1 molecular clones or Tax expression plasmid. After immunoprecipitation of endogenous UPF1, the amount of Gl-PTC mRNA was quantified by qRT-PCR with two different oligonucleotide couples specific either of full-length or only the 3′ portion downstream of PTC. Transfection of the NMD-inhibiting HTLV-1 molecular clone was correlated with a drastic loss of Gl-PTC mRNA associated with UPF1 in comparison to its derivative lacking the ΔpX region (Fig. 3a). The expression of Tax alone also caused a significant decrease in Gl-PTC mRNA amount bound to UPF1 (Fig. 3a). Moreover, since UPF1 nonspecifically binds RNA, the same experiment was carried out using Gl-WT instead of Gl-PTC as the reporter mRNA. Tax was also able to decrease the affinity of UPF1 for Gl-WT mRNA (Supplementary Fig. 3a).

To gain molecular insights into this effect, we performed a series of in vitro tests. First, we performed electrophoretic mobility shift assays (EMSA) by incubating 30-mer radiolabelled single-stranded RNA (ssRNA) with increasing concentrations of purified UPF1-HD in the presence or absence of $Tax_{CACA}$. The presence of Tax markedly reduced the affinity of UPF1 for the ssRNA (Fig. 3b, lanes 6–10) compared with UPF1 alone (Fig. 3b, lanes 1–5). As expected, $Tax_{CACA}$ did not bind the RNA substrate (Fig. 3b, lane 6). To measure the rate constants of the UPF1-HD/RNA interaction in the presence of Tax, we used Bio-Layer Interferometry (BLItz, ForteBio). We immobilised a 5′-biotinylated RNA or DNA 30-mer on a streptavidin-coated biosensor and incubated it with 10 nM of UPF1-HD or UPF1-HD/$Tax_{CACA}$ complex (Supplementary Fig. 3b, c). The biosensor was further washed without protein to monitor passive dissociation. Consistent with previous results, $Tax_{CACA}$ mutant did not bind to the substrates. The obtained sensorgrams showed that the real-time association kinetics (upward slope) of UPF1-HD markedly decreased in the presence of Tax for both substrates (Supplementary Fig. 3b, c), suggesting that Tax prevents the association of UPF1 with nucleic acids. To extrapolate the association ($k_{on}$) and dissociation constants ($k_{off}$) to calculate $K_D$, we performed a BLItz assay using increasing amounts of the UPF1-HD or UPF1-HD/$Tax_{CACA}$ preformed complex (Supplementary Fig. 3d, e). Because of the lower affinity of the complex for DNA compared with UPF1-HD alone, the association time was extended to 600 s. The collected data are summarised in the table of Fig. 3c. As expected, the presence of Tax increased by approximately five times the $K_D$ of UPF1-HD from 8.25 to 40.6 nM. Interestingly, the analysis of $k_{on}$ and $k_{off}$ revealed that, while the $k_{on}$ showed a 10-fold decrease in the presence of Tax ($31.6 \times 10^4$ for HD to $3.05 \times 10^4$ M$^{-1}$ s$^{-1}$ for HD/$Tax_{CACA}$), $k_{off}$ slightly decreased ($2.6 \times 10^{-3}$ for HD to $1.24 \times 10^{-3}$ s$^{-1}$ for HD/$Tax_{CACA}$). Hence, Tax reduced UPF1-HD affinity for its nucleic acid substrate, mainly by preventing its association with nucleic acids.

Finally, to identify the residues critical for Tax binding, we analysed several UPF1-HD mutants. We focused on two of them that were extensively used in the literature and that were associated with RNA affinity and ATP hydrolysis defects: R843C and DE636AA, respectively. R843 is localised just below the CH domain at the RNA entry site and contacts the RNA phosphate backbone[21] (Fig. 3d). The R843C mutation of human UPF1, together with the corresponding yeast R779C mutation, has been shown to confer the strongest NMD inhibition effect in vivo[35,59]. Conversely, the DE636 residues are localised within the ATP

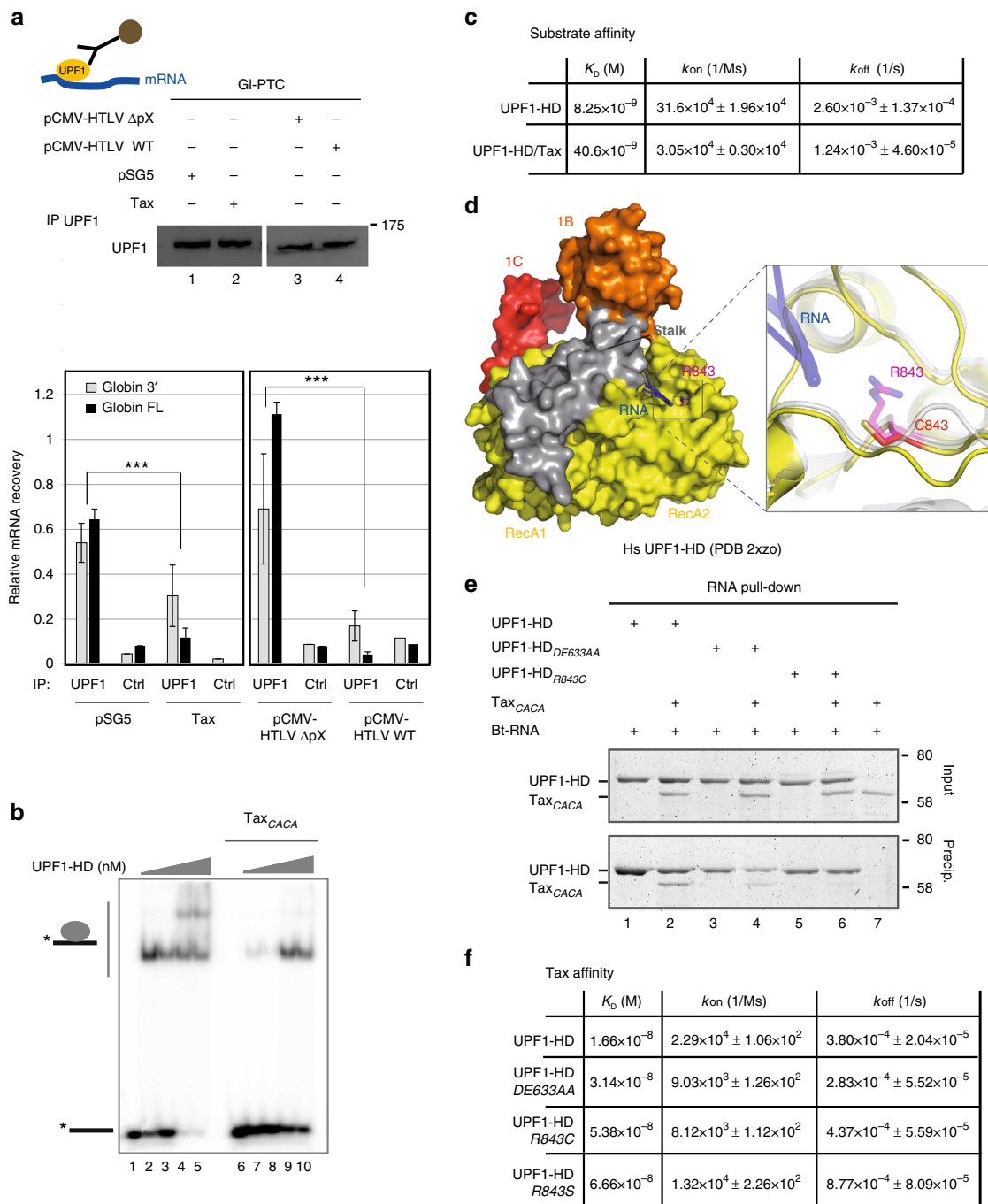

**c** Substrate affinity

| | $K_D$ (M) | $k_{on}$ (1/Ms) | $k_{off}$ (1/s) |
|---|---|---|---|
| UPF1-HD | $8.25 \times 10^{-9}$ | $31.6 \times 10^4 \pm 1.96 \times 10^4$ | $2.60 \times 10^{-3} \pm 1.37 \times 10^{-4}$ |
| UPF1-HD/Tax | $40.6 \times 10^{-9}$ | $3.05 \times 10^4 \pm 0.30 \times 10^4$ | $1.24 \times 10^{-3} \pm 4.60 \times 10^{-5}$ |

Hs UPF1-HD (PDB 2xzo)

**f** Tax affinity

| | $K_D$ (M) | $k_{on}$ (1/Ms) | $k_{off}$ (1/s) |
|---|---|---|---|
| UPF1-HD | $1.66 \times 10^{-8}$ | $2.29 \times 10^4 \pm 1.06 \times 10^2$ | $3.80 \times 10^{-4} \pm 2.04 \times 10^{-5}$ |
| UPF1-HD DE633AA | $3.14 \times 10^{-8}$ | $9.03 \times 10^3 \pm 1.26 \times 10^2$ | $2.83 \times 10^{-4} \pm 5.52 \times 10^{-5}$ |
| UPF1-HD R843C | $5.38 \times 10^{-8}$ | $8.12 \times 10^3 \pm 1.12 \times 10^2$ | $4.37 \times 10^{-4} \pm 5.59 \times 10^{-5}$ |
| UPF1-HD R843S | $6.66 \times 10^{-8}$ | $1.32 \times 10^4 \pm 2.26 \times 10^2$ | $8.77 \times 10^{-4} \pm 8.09 \times 10^{-5}$ |

binding and hydrolysis cleft. The DE636AA mutant has been shown to affect the ATPase activity of UPF1 despite its ability to bind ATP[55]. We produced histidine-tagged UPF1-HD$_{R843C}$ and UPF1-HD$_{DE636AA}$, and we performed an RNA pull-down assay to qualitatively estimate their RNA-binding affinity in the presence of Tax. As expected, the binding of Tax to UPF1-HD induced a reduction of its RNA affinity (Fig. 3e, lanes 1 and 2). The same effect was observed using UPF1-HD$_{DE636AA}$ mutant (Fig. 3e, lanes 3 and 4). Interestingly, UPF1$_{R843C}$ did not interact with Tax in this assay. As a consequence, it did not either show decreased binding to RNA (Fig. 3e, lanes 5 and 6). The interaction between UPF1-HD$_{R843C}$ and RNA was likely due to the numerous residues involved in the RNA interaction within the channel[21]. We

performed BLItz assays to quantitatively assess the Tax-UPF1 affinity using a highly sensitive technique. We immobilised GST-tagged Tax$_{CACA}$ on the sensor and measured the binding parameters of UPF1 WT and mutants (Supplementary Fig. 3f–h).

Consistent with RNA pull-down results, the R843C mutation decreased the dissociation constant ($K_D$) for Tax protein ~3.2 times compared with WT (Fig. 3f and Supplementary Fig. 3f and h). The DE636AA mutant of UPF1 showed slightly affected binding by ~1.8 times compared with WT (Fig. 3f and Supplementary Fig. 3f, g). Unfortunately, we could not affirm that the R843 residue of UPF1 was directly contacted by Tax because C843 possibly formed a disulphide bond with other cysteines affecting the global folding of the HD. To assess whether

Tax bound directly to the R843 residue, we produced histidine-tagged UPF1-HD$_{R843S}$ and performed a BLItz assay to test the interaction with Tax (Supplementary Fig. 3f and i).The UPF1-HD$_{R843S}$ mutant showed four times less affinity for Tax compared with WT. That finding is consistent with the hypothesis of a direct interaction of Tax with R843 of the HD.

Considering the position of the R843 residue (Fig. 3d), these results strongly suggest that Tax may interfere with the entry of RNA into the binding channel of UPF1, preventing its association with, rather than dissociation from, the substrate.

**Tax binds UPF1 containing RNP and modifies UPF1 behaviour.** However, as observed in Fig. 3c, the in vitro analysis showed that the $K_D$ of the UPF1-HD/Tax complex, although higher than that of UPF1-HD alone, was still rather low, suggesting that a significant portion of the UPF1/Tax complex was nonetheless associated to the substrate. This was supported by RNA pull-down experiments' observation (Fig. 3e, lane 2).

To verify the existence of a UPF1/Tax complex bound to NMD substrates and to assess its physiological relevance and whether UPF1 was still active under these circumstances, we immuno-precipitated transiently expressed Tax protein in HeLa cells and quantified the associated β-globin mRNA as described previously (Fig. 4a). The antibody against Tax was able to precipitate approximately ten times and five times more Gl-PTC mRNA compared with the control immunoglobulin and Tax-free samples, respectively. The association of Tax with Gl-PTC mRNA was also four times stronger than Gl-WT. Interestingly, after treatment of HeLa cells with UPF1 siRNAs, three times less Gl-PTC mRNA was co-immunoprecipitated with Tax compared with the control (Fig. 4a). These data were in agreement with the visualisation of semiquantitative RT-PCR amplicons after migration through an agarose gel (Supplementary Fig. 4a). Similar experiments were carried out with the HTLV-1 molecular clones WT and ΔpX, and showed the same trend (Supplementary Fig. 4b). To confirm the concomitant presence of UPF1 and Tax on NMD substrates, we transfected HeLa cells with HA-UPF1 together with Tax expression plasmids to perform a double-RIP experiment (Fig. 4b). The hemagglutinin (HA) tag was used as bait to immunoprecipitate UPF1 (Fig. 4c, lane 6). The precipitate was incubated with an anti-Tax antibody to further pull down the UPF1-HD/Tax$_{CACA}$ complexes. The western blot showed the presence of both Tax and UPF1 in the sample after the second immunoprecipitation (Fig. 4c, lane 7). Concomitantly, mRNAs

were extracted from fractions of the immunoprecipitates, and the presence of Gl-PTC mRNA was analysed by semiquantitative RT-PCR (Fig. 4d, lanes 4 and 5). Notably, the amount of Gl-PTC mRNA in the second immunoprecipitate represented a fraction of the total transcript associated with UPF1. Taken together, these data demonstrated that, although to a lesser extent, UPF1 was still able to bind NMD targets in the presence of Tax ex vivo.

With the aim to confirm these data, we monitored Tax binding to UPF1-HD already associated with the DNA substrate by using the BLItz assay (Fig. 4e). We first measured the association of UPF1-HD to a biotinylated 30-mer DNA oligonucleotide bound to the sensor tip (Fig. 4e, both curves up to 300 s), and then we added or did not add Tax$_{CACA}$ in *trans* at 330 s (Fig. 4e, continuous blue line and red dashed line, respectively). The increasing BLI signal in the sensorgram indicated the association kinetics of Tax on UPF1-HD instead of the dissociation of the UPF1-HD/Tax complex. Next, we wondered whether Tax could modify the UPF1-HD behaviour on its substrate. It has been shown that the binding of ATP or its non-hydrolysable analogue (ADPNP) within the active site modulates the nucleic-acid affinity of UPF1, most likely at the base of enzyme translocation[20,21,33,55]. In general, the presence of ATP or ADPNP decreases UPF1 affinity for the substrate. Hence, we performed a BLItz assay using the UPF1-HD and UPF1-HD/Tax$_{CACA}$ preformed complex to test the effect of the nucleotide presence. According to the observed $k_{on}$ (Fig. 3c), we used seven times more complex than UPF1 protein alone. The presence of ATP or ADPNP strongly reduced UPF1-HD affinity for the ligand, while a weaker effect was observed using the UPF1-HD/TAX$_{CACA}$ complex (Fig. 4f).

These results demonstrate that Tax can be embedded within UPF1-containing RNP and that, under these conditions, UPF1-HD is less sensitive to the presence of ATP.

**UPF1 translocation is inhibited by Tax binding.** Finally, we used magnetic tweezers (MT, Picotwist®) to determine the real-time effect of Tax on UPF1-HD protein actively running on DNA. The experimental configuration is a 1200-base pair (bp) DNA hairpin tethered between a glass surface and a magnetic bead[33,60]. The biotinylated 5′ end of DNA binds the streptavidin-coated bead, while the digoxygenated 3′ end interacts with the antidigoxygenin antibody attached to a glass surface. A controlled force was applied to the ends of the hairpin using two magnets (Fig. 5a), and $Z(t)$ was measured by tracking the position of the

**Fig. 3** Tax binding near the entry site of RNA decreases UPF1 affinity for its substrate. **a** The upper panel shows the levels of endogenous UPF1 immunoprecipitation (IP) from mock (lane 1), Tax (lane 2), pCMV- HTLV-1 ΔpX (lane 3) and pCMV- HTLV-1 WT (lane 4)-transfected HeLa cells. The lower panel represents the relative quantification of Gl-PTC RNA recovery upon UPF1 RIP. The ΔΔCt method of mRNA quantification was applied to each experimental condition. The control sample (Ctrl) derived from IP using immunoglobulin G (IgG) antibody. The values represented in each graph correspond to the mean of at least three biological replicates, and the error bars correspond to the SD. ***$P < 0.005$ with Student's $t$-test (two-tailed, unpaired). **b** Representative native 8% polyacrylamide gel illustrating the interaction of UPF1-HD protein with a 30mer-ssRNA (grey line) labelled with $^{32}$P (black star) with or without Tax$_{CACA}$ factor. The RNA substrate (1 nM) was incubated with increasing concentrations of UPF1-HD (0, 1, 3, 5 and 10 nM) alone or with 300 nM of Tax$_{CACA}$. The absence of an interaction between Tax$_{CACA}$ and the substrate was also verified (lane 6). **c** Recapitulative tables of Bio-Layer Interferometry experiments listing $K_D$, $k_{on}$ and $k_{off}$ in real-time measurements of UPF1-HD and UPF1-HD/Tax complex binding to 5′ biotinylated 30mer-ssDNA. These data were obtained using the BLItz® System instrument and BLItz ®Pro 1.2 software (ForteBio). **d** Crystal structure of human UPF1-HD complexed with RNA substrate showing the position of the R843 residue[21]. The magnification shows the structural model of the R843C mutant produced using I-Tasser software. **e** Protein co-precipitation with 3′ end-biotinylated 30-mer ssRNA (Bt-RNA). Combinations of UPF1-HD (lanes 1 and 2), UPF1-HD$_{DE633AA}$ (lanes 3 and 4) or UPF1-HD$_{R843C}$ (lanes 5 and 6) were mixed with Tax$_{CACA}$ (lanes 2, 4 and 6) and incubated in a buffer containing 200 mM NaCl before co-precipitation. Tax$_{CACA}$ was incubated with Bt-RNA alone as a control for aspecific binding (lane 7). Input (20% of total) and pull-down fractions were analysed by 12% SDS-PAGE followed by coomassie blue staining. **f** Recapitulative tables of BLItz listing $K_D$, $k_{on}$ and $k_{off}$ in real-time measurements for UPF1-HD$_{WT}$, UPF1-HD$_{DE633AA}$, UPF1-HD$_{R843C}$ and UPF1-HD$_{R843S}$ binding to Tax. Uncropped scans related to Fig. 3 are available in Supplementary Fig. 7

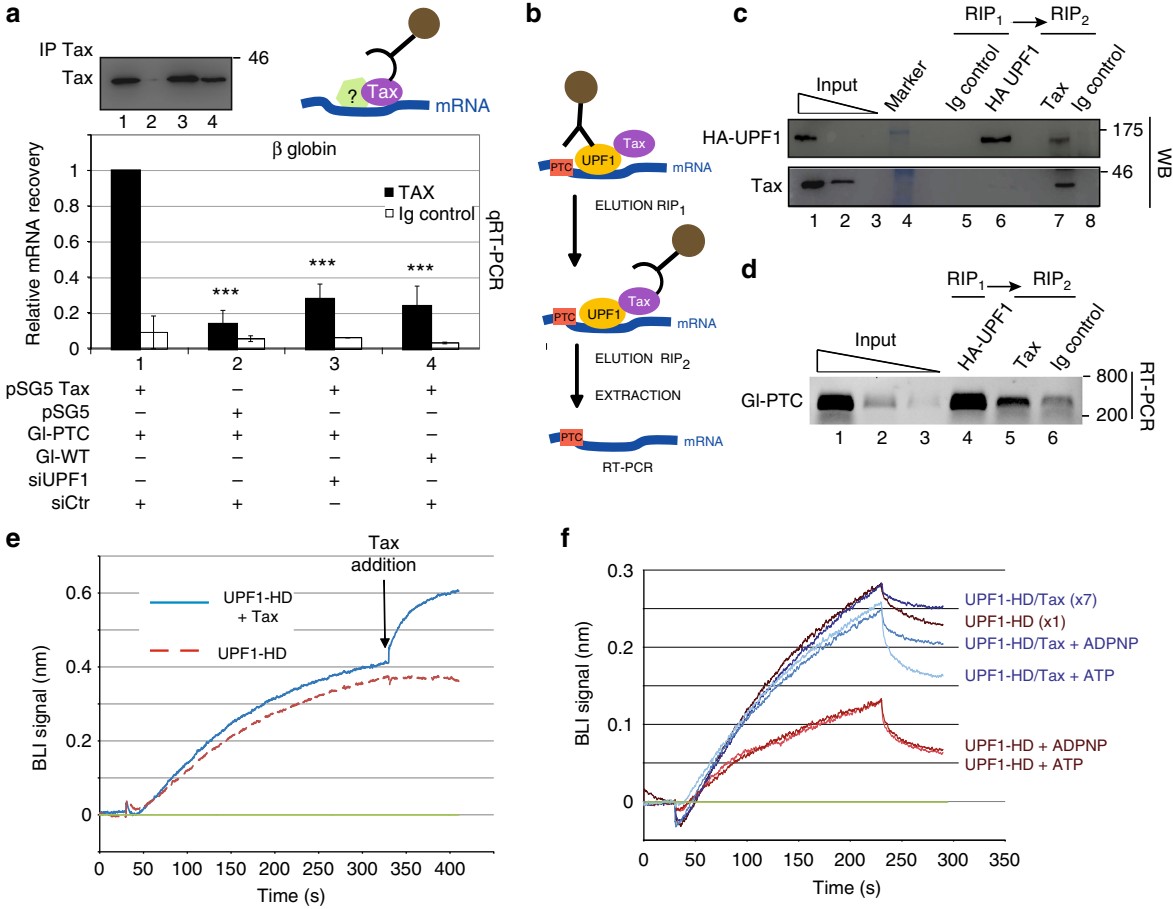

**Fig. 4** The UPF1-HD/Tax complex shows residual RNA-binding ability. **a** Quantification of precipitated GI-PTC from Tax immunoprecipitation (black bars) vs. IgG immunoprecipitation control (white bars) using the globin 3′ oligos. The histogram represents the quantification of relative RNA recovery upon normalisation to input RNA under condition 1 that was set to 1. The ΔΔCt method of mRNA quantification was applied to each experimental condition. The construct encoding Tax (conditions 1, 3 and 4) or empty vector (lane 2) were co-transfected with reporter plasmids expressing GI-PTC (lanes 1–3) or GI-WT (lane 4). The cells were treated with non-targeting siRNA (siCtr; lanes 1, 2 and 4) or UPF1 siRNA (siUPF1; lane 3). The values represented in each graph correspond to the mean of at least three biological replicates, and the error bars correspond to the SD. ***$P < 0.005$ with Student's $t$-test (two-tailed, unpaired). Immunoprecipitated Tax was analysed by western blotting using anti-Tax antibody (upper left panel). **b** Schematic representation of the double-RIP workflow. **c** Western blot analysis monitoring the presence of both UPF1 and Tax in each fraction. **d** Agarose gel showing the products of RT-PCR using globin 3′ oligonucleotides with input and immunoprecipitated RNA samples. **e** Real-time sensorgram of the Bio-Layer interferometry experiment showing UPF1-HD binding to 5′-biotinylated 30mer-DNA (red dashed line), and the interaction of UPF1-HD with Tax_{CACA} when Tax_{CACA} was added at 330 s (blue line). **f** Association and dissociation curves of UPF1-HD (10 nM) and UPF1-HD/Tax (70 nM) preformed complex to the DNA-coated sensor with or without ATP or ADPNP. Given that the $k_{on}$ of the UPF1-HD/Tax_{CACA} complex is one order of magnitude lower than that UPF1-HD, the concentration of the protein complex was adjusted to have almost the same association kinetics with DNA than the enzyme alone. Biosensor DNA tips were loaded with: UPF1-HD (brown), UPF1-HD supplemented with 2 mM of ADPNP (red) or 2 mM of ATP (orange), UPF1-HD/Tax (dark blue), UPF1-HD/Tax supplemented with 2 mM of ADPNP (middle blue) or 2 mM of ATP (light blue). Uncropped scans related to Fig. 4 are available in Supplementary Fig. 8

magnetic bead in real-time. During the helicase MT assay, the unwinding activity of UPF1-HD was deduced from changes in the extension of the DNA molecule. As expected, we observed a saw-tooth track of unwound bases as a function of time. The rising edge corresponds to the hairpin unwinding by UPF1-HD. The falling edge represents hairpin refolding immediately behind UPF1-HD that is translocating on single-stranded (ss) DNA (Fig. 5b; Supplementary Fig. 5a and ref. [33]). Consistently with the previously published data, the average unwinding and ss-translocation rates of UPF1-HD were 0.58 and 1.40 bp.s⁻¹, respectively, for this experiment[33]. To assess the effect of Tax binding, we performed a MT assay in which actively unwinding/translocating UPF1 was identified and monitored before addition of Tax into the reaction chamber. During the helicase MT assay,

two injections of Tax_{CACA} were performed between 1150–1325 and 3100–3275 s in the experiment presented in Fig. 5c. We observed 51 unwinding events out of 63 in which the UPF1-HD activity was affected. Among a whole series of unwinding events affected by Tax, we observed instantaneous refolding of the hairpin, indicating the dissociation of UPF1-HD from the substrate (Fig. 5c, upper track at 3625 s; Supplementary Fig. 5e, f). When Tax interacts with UPF1-HD that is undergoing active ss-translocation, the effect is more complex, as shown in Fig. 5c, lower track and Supplementary Fig. 5b–d. In these panels, the recorded tracks illustrate all three types of inhibiting events affecting the UPF1 translocation. They present a characteristic burst shape with a normal rising edge compared with the UPF1-HD recorded track (Fig. 5b and Supplementary Fig. 5a). However,

their falling edge is interrupted in almost 44% of the total Tax interacting events, and the activity of ss-translocation of UPF1 is blocked (Fig. 5c, lower track, from 1920 to 2370 s and from 2414 to 3660 s). In 41% of the events, the hairpin instantaneously refolds (Fig. 5c, lower track, at 3660 s), indicating that Tax induces the dissociation of UPF1-HD from its substrate. In 15% of the inhibited events, we observed a rapid refolding of the hairpin that we call sliding, followed by a blocking event (Fig. 5c, lower track, between 2372 and 2417 s; Supplementary Fig. 5b). This temporary sliding of the UPF1/Tax complex was likely boosted by the hairpin rezipping behind UPF1. These data demonstrate that Tax destabilises UPF1 during both unwinding and translocation, leading to its dissociation from the substrate.

## Discussion

UPF1 ATPase activity plays a key role during NMD, while the biological meaning of UPF1 translocation remains unclear. The helicase and ATP binding and hydrolysis activities were linked to a selective association with NMD targets[34,61], as well as to the proper translation termination at PTC and overall nonsense mRNP remodelling[30,34,62]. In this study, we describe how the HTLV-1 Tax factor inhibits NMD during retroviral infection by affecting UPF1 activity at several levels, disturbing substrate binding and translocation.

Our earlier work showed that the viral protein Tax interacts with NMD components, including UPF1, UPF2 and the translation initiation factor INT6/eIF3E, inhibiting the degradation of some cellular and viral mRNAs by NMD[41,63]. Here, we strengthened those results by analysing the effect of Tax on several host-endogenous NMD targets presenting different NMD-triggering features and in the context of an HTLV-1 molecular clone mimicking HTLV-1 infection (Figs. 1a–f). On the basis of these results, Tax should be considered a regulator of global host gene expression during HTLV-1 infection.

We established the direct interaction of Tax with UPF1, through its helicase core and, to a lesser extent, the N-terminal CH domain (Figs. 2b, c). Since UPF1 is a very tightly regulated enzyme[20–22,33], we investigated the possible Tax-mediated modulation of its activities. While UPF1 is a nucleic-acid-dependent hydrolase, we characterised two different ways in which Tax deregulates UPF1 functioning, depending on when their interaction takes place.

The first level of regulation involves the interaction with Tax before UPF1 binding to the nucleic-acid substrate. We found that the presence of Tax was associated with a 10 times lower affinity of UPF1 for nucleic acids (Figs. 3a–c). Measurement of the UPF1/Tax complex affinity ($k_{on}$ and $k_{off}$) showed that the UPF1 association with RNA, instead of its dissociation, is affected. This Tax-mediated loss of RNA affinity is most likely the cause of the low ATP hydrolysis rate of the enzyme (Figs. 2d, e). We observed that UPF1 mutations of the R843 residue (R843C or R843S) decrease the Tax interaction (Figs. 3e, f). On the basis of the structural information, R843 has been observed to interact with the phosphate backbone of the RNA 5′ extremity and is localised within the nucleic-acid-binding channel in an external region exposed to the solvent[21] (Fig. 3d). In our hands, the R843C mutation only slightly decreased the affinity of UPF1-HD for RNA in vitro, most likely due to compensatory effects exerted by other residues of the channel. The localisation of R843 strongly suggests that the RNA

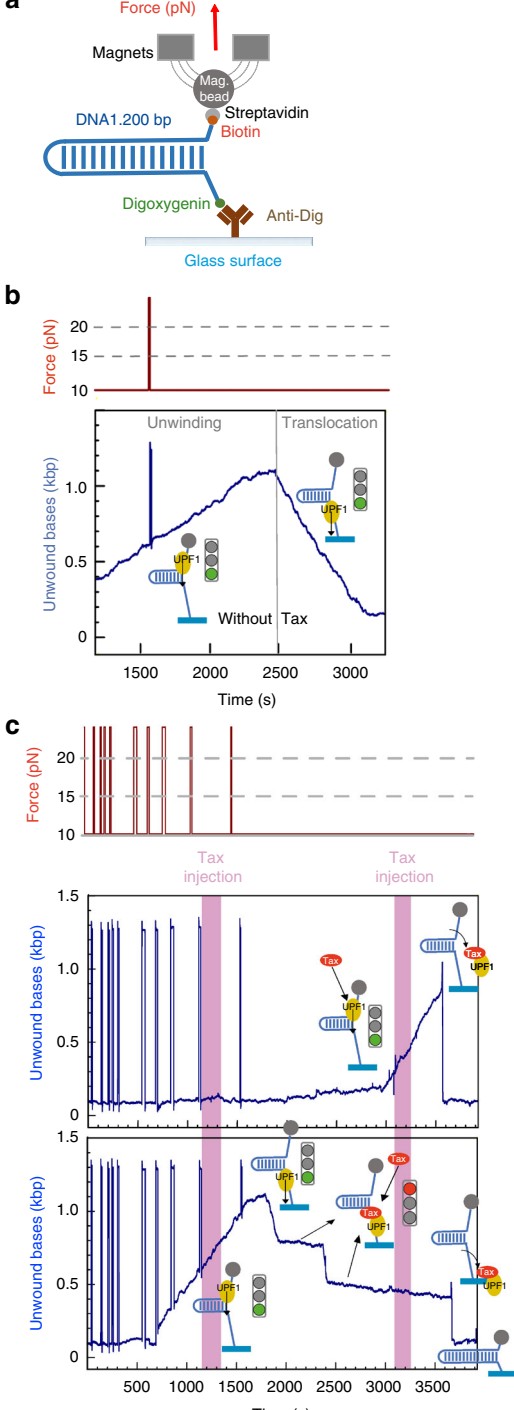

**Fig. 5** Tax blocks UPF1 translocation. **a** Schematic representation of the DNA substrate used for the magnetic tweezers' set-up. **b** Experimental magnetic tweezer traces showing the activity of UPF1-HD under a saturating concentration of ATP (blue line). The trace showing the force monitoring (red line) is shown above the recorded track. The number of unwound bases is deduced from the molecular extension Z(t) obtained at F = 10 piconewton (pN). From 1000 to 2500 s, the helicase unwound the ~1200 bp DNA hairpin. From 2500 to 3200 s, the DNA hairpin refolded while the UPF1-HD translocated on the ss-DNA. **c** The upper and lower panels show two types of enzymatic activity detected for UPF1-HD in the presence of Tax. On the upper panel, UPF1-HD is unwinding the DNA hairpin between 3000 and 3625 s before dissociation induced by Tax binding. On the lower panel, UPF1-HD is unwinding the whole DNA hairpin between 500 and 1750 s and translocating between 1750 and 1920 s on the ss-DNA. Tax blocks UPF1-HD translocation twice between 1920 and 2370 s and 2414 and 3660 s. A sliding event and the final dissociation of UPF1-HD can be observed at 2372 and 3660 s, respectively

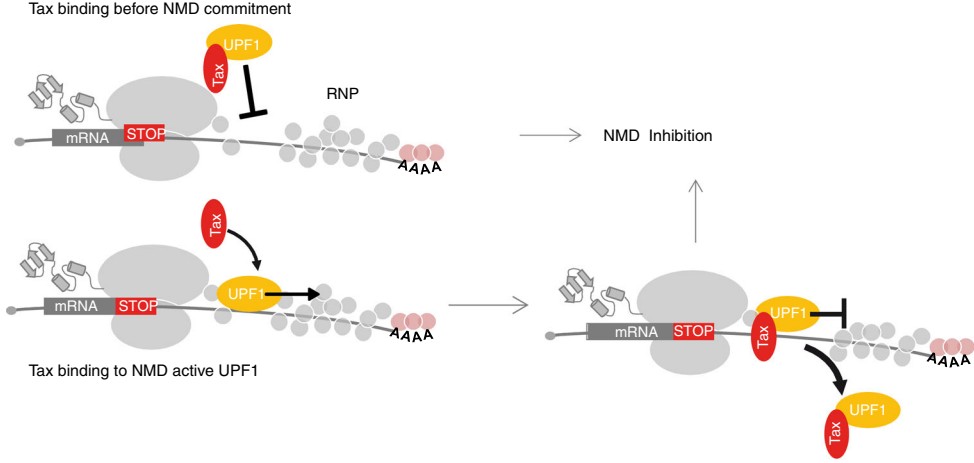

**Fig. 6** Model proposed for Tax-mediated inhibition of UPF1 activity. The fate of UPF1 depends on the time frame of interaction with Tax that occurs during HTLV-1 infection. Tax can prevent the recruitment of UPF1 to the RNA substrate. Alternately, Tax can affect the activity of previously bound RNA and NMD-engaged UPF1: the translocation activities of UPF1 can be blocked by Tax binding. The UPF1/Tax complex then dissociates from the RNA substrate or remains stuck in P-bodies

entry site is sterically hindered by Tax bound to the UPF1 helicase core domain.

The second level of UPF1 enzymatic activity inhibition involves the interaction with Tax in a sequential manner, after UPF1 is already bound to its nucleic-acid substrate. Our results showed that a significant fraction of the UPF1/Tax complex is associated with nucleic acids in vitro and ex vivo, supporting the relevance of this complex to RNA (Figs. 3 and 4). To understand the impact of Tax in this configuration of events, we monitored an actively running UPF1 at the single-molecule level (Fig. 5). Recently, the striking processivity and single-stranded translocation activity of UPF1 were formally demonstrated[33] (Fig. 5a and Supplementary Fig. 5a). The enzymatic features of UPF1 are strongly linked ex vivo to mRNP disassembly, a mandatory step for mRNA degradation during NMD[7]. Translocation ability may also be included in these essential UPF1 activities. In vitro, the affinity of UPF1 for nucleic acids is so high that under conditions that dissociate proteins strongly bound to DNA, UPF1 still translocates[33]. Surprisingly, Tax was able to induce UPF1 dissociation during double-stranded DNA unwinding (Fig. 5c and Supplementary Fig. 5e–f). Moreover, the effect of Tax differed on UPF1 that was actively translocating on single-stranded DNA. It mainly blocked UPF1 translocation for a while before inducing its dissociation from the substrate under the pressure of hairpin refolding (Fig. 5c and Supplementary Fig. 5b–d). This Tax-mediated effect should also be related to the accessibility of the Tax-binding site. In fact, during UPF1 translocation from the 5′ to 3′ end of DNA, the refolding hairpin is likely localised on its backside close to R843 (Fig. 3d). The access of Tax may be sterically hindered by DNA, and it may pass through a blocked UPF1 conformation before reaching a dissociation conformation. Moreover, interferometry observations showing that the UPF1/Tax complex was insensitive to the presence of ATP or ATP analogues in the binding site (Figs. 4f and 2e), combined with the blocking effect observed at a single-molecule level (Fig. 5c and Supplementary Fig. 5b–d), suggest that Tax likely stabilises a "closed" conformation of UPF1 that is reminiscence of the CH-domain effect. In fact, structural analysis showed that the CH domain stabilises a clamping conformation of UPF1-HD by narrowing the groove around RNA and by preventing translocation of the enzyme[21,33]. Furthermore, the slight interaction of

Tax with the CH domain (Fig. 1b) might suggest a possible concerted inhibition mechanism of UPF1 enzymatic activity.

In our model, HTLV-1 Tax showed two mechanisms of action, depending on whether UPF1 was already bound to mRNA substrate (Fig. 6). Physiologically, this dual inhibition mechanism suggests that Tax may inhibit NMD during early steps before UPF1 is engaged with the substrate, as well as in later steps once UPF1 translocation is triggered. In the latter case, analogously to UPF1 ATP binding and hydrolysis mutations[7], Tax might prevent mRNP disassembly and completion of NMD factor turnover necessary for correct mRNA decay. Consistently, in the presence of Tax, aberrant P-body profiles and higher levels of phosphorylated UPF1 were also observed (Supplementary Fig. 1 and ref. [41]). As a consequence, the activity of Tax may be widespread by regulating the expression of several host and pathogen mRNAs that are usually degraded by the NMD machinery, including cellular mRNA, such as GADD45α and viral mRNAs (Fig. 1f and refs. [5,6]). Tax-mediated regulation may offer advantages to HTLV-1, not only by ensuring the expression of its own genome but also by controlling several cellular functions that potentially participate in leukemogenesis. Thus, as Tax is known to have a mutagenic effect, blocking of NMD is likely to facilitate expression of truncated or mutated proteins that exert deleterious effects.

In conclusion, we establish that UPF1 translocation jamming correlates with strong NMD inhibition, supporting the notion that UPF1 translocation plays a key role during NMD. Analysis of specific UPF1 mutants in the absence of an exogenous factor will be suitable to identify a more direct relationship between translocation and NMD. Moreover, it will also be interesting to further describe the interplay of Tax with other NMD factors that contribute to the complex intramolecular regulation of UPF1.

## Methods

**Plasmids**. All directed mutagenesis experiments were carried out by a 20-cycle PCR using Phusion polymerase (Thermo Fisher Scientific), a pair of reverse complementary primers (2 μM) and 50 ng of parental plasmid. PCR products were digested with DpnI restriction enzyme and used for XL1blue transformation. The pGEX2T-Tax$_{WT}$[57] was mutated into pGEX2T-Tax$_{CACA}$ using 5′-GTTTGGAGACgctGTACAAGGCGACTGGgccCCCATCTCTGGG-3′ oligonucleotide. UPF1 full-length and domains were produced with the pHL UPF1-FL, pHL UPF1-HD, pHL UPF1-CH-HD and pHL UPF1-HD-SQ constructs previously described[22]. UPF1-HD$_{DEAA}$ and UPF1-HD$_{R843C}$ were modified using primers 5′-GCTCCATTTTAATCGcCGcA AGCACCCAGGCCACC-3′ and 5′-

CTGTCCTGTGTGTGtGcGCCAACGAGCACCAAG-3′, respectively. Viral molecular pCMV HTLV-1 WT[64] was modified into pCMV HTLV ΔpX by digestion/religation of the two BlpI sites within the Tax/Rex-coding sequence.

**Antibodies**. The antibodies used were mouse monoclonal antibody against Tax (clone 474, Covalab, 1:1000), mouse monoclonal antibodies against HA (monoclonal clone 7, Sigma-Aldrich, 1:1000), rabbit polyclonal against UPF1 (rabbit polyconal clone A301-902A, Bethyl, 1:20,000), Anti α-His (polyclonal, Abcam ab18184, 1:1000) and anti β-beta actin (monoclonal AC15, Sigma-Aldrich, 1:5000).

**Protein expression and purification**. GST-Tax proteins were produced from BL21-DE3 bacteria (Novagen) induced with 0.1 mM IPTG at 25 °C during 1 h. After sonication in MTPBS (150 mM NaCl, 12.5 mM $Na_2HPO_4$, 2.5 mM $KH_2PO_4$, 100 mM EDTA pH 7.3, 0.05% Triton,10% glycerol), the lysate was incubated for 3 h at 4 °C with glutathione magnetic beads (Promega). The beads were washed three times, and Tax was further eluted in three fractions with MTPBS supplemented with 10 mM reduced glutathione at 4 °C. HisUPF1 proteins were produced and purified as previously described[22]. Briefly, BL21 DE3 bacteria were transformed and induced overnight at 16 °C in 1 l of LB. Lysis was carried by sonication in 1.5× phosphate-buffered saline (PBS), 0.1% NP40, 20 mM imidazole, 1 mM magnesium acetate and 10% glycerol with lysozyme. Soluble lysate was applied to 1 ml of NiNTA beads (Macherey-Nagel) for 3 h at 4 °C. The NiNTA resin was washed three times, and UPF1 proteins were eluted in three fractions with lysis buffer supplemented with 150 mM imidazole. The Tax/UPF1-HD complex was purified with a NGS HPLC system (Bio-Rad) after co-lysis of 200 ml of HisUPF1-HD and 1 l of Tax-expressing bacteria pellets. Lysis was performed in MTPBS with lysozyme and 5 mM imidazole. The co-lysate was first applied to a HisTrap column. The eluate (250 mM Imidazole) was dialysed in buffer PBS (1 × PBS, 10 % glycerol, 4 μM $MgCl_2$, 6 μM $ZnCl_2$, 0.1% NP40) and further purified with a GST-Trap column. The Tax/UPF1-HD complex was finally eluted with PBS buffer supplemented with 10 mM reduced glutathione and dialysed against PBS buffer.

**RNA and protein co-precipitation**. The RNA co-precipitation and GST pulldown were performed as described previously[20,22,41,65]. Briefly, in the RNA co-precipitation assay, proteins or preformed protein complex (2 μg) were mixed with 70 pmoles of 3′ end-biotinylated ssRNA (CGUCCAUCUGGUCAUCUAGU-GAUAUCAUCG[BtnTg]) in binding buffer (20 mM HEPES pH 7.5, 150 mM potassium acetate, 2 mM magnesium acetate, 1 mM dithiothreitol (DTT), 6.3% (v/v) glycerol and 0.1% (w/v) NP-40). The reactions were performed in a final volume of 30 μl and incubated for 20 min at 30 °C. Then, 5 μl of pre-coated streptavidin-coupled magnetic beads (Dynabeads, Life Technologies) were added before further incubation for 1 h at 4 °C. Unless indicated otherwise, the beads were washed with binding buffer containing 200 mM NaCl. Proteins were eluted by addition of 7.5 μl of SDS loading buffer directly to the beads. The various fractions were subsequently analysed by 12% SDS-PAGE. For precipitation of protein complexes by GST-bait protein, the magnetic beads were replaced with 12 μl of GST resin (50% slurry, Promega). The resin was washed three times with 500 μl of binding buffer containing 200 mM NaCl and eluted with SDS loading buffer. Eluates were separated by 10% SDS-PAGE and visualised by coomassie staining.

**ATP binding**. Equilibrated amounts of proteins were applied to a nitrocellulose membrane with a slotblot apparatus. The membrane was soaked in blocking buffer (20 mM HEPES pH 7.0, 50 mM potassium acetate, 2.5 mM magnesium acetate, 2 mM DTT, 3% BSA (w/v), 10% (v/v) glycerol) and incubated on a rocking platform for 1 h at room temperature. The blocking buffer was then replaced with binding buffer (20 mM HEPES pH 7.0, 50 mM potassium acetate, 2.5 mM magnesium acetate, 2 mM DTT, 1.5% (w/v) BSA, 10% (v/v) glycerol) supplemented with 30 mCi of $[α-^{32}P]$-ATP (Perkin Elmer) before further incubation for 20 min at room temperature. The membrane was washed twice with blocking buffer before being dried and analysed by phosphorimaging.

**ATP hydrolysis**. ATPase assays were carried out as described in ref. [22]. Briefly, 10 pmol of UPF1 protein and UPF1/Tax preformed complex were incubated at 30 °C in a 10-μl reaction mixture containing 20 mM MES pH 6.0, 100 mM potassium acetate, 1 mM DTT, 0.1 mM EDTA, 1 mM magnesium acetate, 1 mM zinc sulphate, 5% (v/v) glycerol, 2 μCi of $[α-^{32}P]$-ATP (800 Ci.mmol$^{-1}$, Perkin Elmer), 25 mM cold ATP and 20 μg.ml$^{-1}$ tRNA. At the indicated times, 2 μl reaction aliquots were withdrawn and quenched with 10 mM EDTA and 0.5% (v/v) SDS. Samples were analysed by phosphorimaging after TLC on polyethyleneimine cellulose plates (Merck) with 0.35 M potassium phosphate (pH 7.5) as migration buffer.

**Electrophoretic mobility shift assay**. The EMSA was performed as described in Fiorini et al[22]. Samples were prepared by mixing a radiolabelled 30-mer oligoribonucleotide (1 nM; CGUCCAUCUGGUCAUCUAGUGAUAUCAUCG) with UPF1 protein (0, 1, 3, 5, 10 nM) with or without $Tax_{CACA}$ (300 nM) in a buffer containing 20 mM MES pH 6.0, 150 mM potassium acetate, 2 mM DTT, 0.2 μg.μl$^{-1}$ BSA and 6% (v/v) glycerol. The samples were incubated at 30 °C for 20 min before

being resolved by native 6.5% polyacrylamide (19:1) gel electrophoresis and analysed by phosphorimaging.

**RNA decay assays and qRT-PCR/RT-PCR**. RNA decay assays were performed to assess the stability of mRNA expressed from a β-globin reporter minigene that was either WT (Gl-WT) or with a PTC in the second exon (Gl-PTC)[41]. For this procedure, 0.5 μg of Gl-PTC or 0.05 μg of Gl-WT constructs were co-transfected as indicated with 0.5 μg of renilla-expressing vector in $0.7 × 10^6$ HeLa cells with jetprime reagent (polyplus transfection). Additional plasmids were co-transfected as indicated in the figures. The medium was changed after 12 h, and cells were further cultivated for 24 additional hours. Then, following RNA decay, the cells were cultivated for 0, 1, 3 or 4 h under DRB treatment (100 μg.ml$^{-1}$) to block transcription. Total mRNAs were extracted using the Macherey-Nagel RNA easy extraction kit and quantified by qRT-PCR using the QuantiTect SYBR Green qRT-PCR kit (Qiagen) and appropriate primers:

GLOBIN (Globin 3′ forward 5′-TTGGGGATCTGTCCACTCC-3′, Globin 3′ reverse 5′-CACACCAGCCACCACTTTC-3′, Globin full-length forward 5′-GATG AAGTTGGTGGTGAGGC-3′, Globin full-length reverse 5′-AGTGATACTTGTG GGCCAGG-3′), *GADD45α* (forward 5′-ACGAGGACGACGACAGAGAT-3′, reverse 5′-GCAGGATCCTTCCATTGAGA-3′), MAP3K14 (forward 5′-TCAGT GCAGAACCAGGTCAG-3′, reverse (5′-GGGGACTGAGAACCACTTCA-3′) and SMG5 (forward (5′-ACAGAATGGGATGCCAGGAA-3′, reverse 5′-TCAACAC TCCAAAAGCCAGC-3′). Normalisation was carried out with respect to renilla mRNA (renilla forward primer 5′-CTAACCTCGCCCTTCTCCTT-3′, renilla reverse 5′-TCGTCCATGCTGAGAGTGTC-3′). The values represented in the graphs correspond to the mean of at least three biological replicates, and the error bars correspond to the SD. Half-lives were calculated for each replicate, and $P$ values were calculated by performing a Student's $t$-test (unpaired, two-tailed) ns: $P$ > 0.05; *$P$ < 0.05; **$P$ < 0.01.

**RNA immunoprecipitation**. For UPF1 RIP, ~$2 × 10^6$ HeLa cells were transfected as described in the figures, harvested and resuspended in lysis buffer (50 mM Tris–Cl, pH 7.5, 1% NP-40, 0.5% sodium deoxycholate, 0.05% SDS, 1 mM EDTA, 150 mM NaCl, protease inhibitor (Roche) and RNAsin (Promega)). Extracts obtained after centrifugation at 12,000$g$ for 15 min were incubated with primary antibody overnight at 4 °C. Protein A and G magnetic beads (dynabeads, Life Technology; 5 μl each) were mixed and coated with PBS + 5% BSA, supplemented with tRNA and RNAsin overnight. After re-equilibration in lysis buffer (supplemented with tRNA), the beads were added to the lysate for 2 h at 4 °C before extensive washing in lysis buffer. The beads were resuspended in elution buffer (50 mM Tris–HCl, pH 7.0. 1 mM EDTA, 10 mM DTT and 1% SDS).

For RIP Tax, we followed the protocol described by Niranjanakumari et al.[66] to perform reversible crosslinking combined with RIP. For each RIP condition, ~$2 × 10^6$ HeLa cells were transfected as described in the figures, harvested and fixed with 0.05% formaldehyde for 20 min at room temperature. Then, 0.25 M glycine was added for 5 min before PBS washing. The cell pellet was resuspended in 2 ml of lysis buffer, and the lysate was sonicated (bioruptor, diagenode). The immunoprecipitation steps were carried out as described for UPF1 RIP, except that extensive washings were performed with 50 mM Tris–Cl, pH 7.5, 1% NP-40, 0.5% sodium deoxycholate, 0.05% SDS, 1 mM EDTA, 500 mM NaCl and 1 M urea. The beads were resuspended in elution buffer and incubated at 70 °C for 45 min for reverse-crosslinking. In the case of the double-RIP experiment, ~$9 × 10^6$ HeLa cells were transfected with 10 μg of Tax and 15 μg of the HA-UPF1 expression plasmids[41], as well as 10 μg of Gl-PTC construct. The cells were treated using similar conditions as described for Tax RIP. First, the HA tag was immunoprecipitated. Elution was carried out at 4 °C in 10 μl of 50 mM Tris HCl, 1 mM EDTA, RNAsin and 20 mM DTT for 1 h, plus one additional hour in 150 μl of 50 mM Tris HCl, 1 mM EDTA complemented with 100 μg.ml$^{-1}$ HA elution peptide. Finally, the elution volume was increased to 500 μl (50 mM Tris–Cl, pH 7.5, 1% NP-40, 0.5% sodium deoxycholate, 0.05% SDS, 1 mM EDTA, 150 mM NaCl, protease inhibitor (Roche) and RNAsin), and Tax immunoprecipitation was performed. After extensive washing, the beads were resuspended in elution buffer and incubated at 70 °C for 45 min for reverse-crosslinking.

For all RIP, RIP and double-RIP experiments, the precipitated RNA was further extracted with RNAzolRT reagent (MRC) and subjected to qRT-PCR.

**DNA and siRNA transfection**. DNA and short interfering RNA (siRNA) co-transfection in HeLa cells was carried out using jetprime® reagent according to the manufacturer's instructions (Polyplus transfection SA). The siCtrl (MISSION siRNA Universal negative control, SIGMA) and siUPF1 (5′-GAUGCA-GUUCCGCUCCAUUGAUGCAGUUCCGCUCCAUU-3′) were used at a final concentration of 10 nM. Twenty-four hours later, the cells were washed with PBS and reincubated in fresh medium for 48 h before harvest.

**Biolayer interferometry**. Biolayer interferometry experiments were carried out using a BLItz apparatus (ForteBio). The affinity of UPF1 HD for its substrate was analysed with a SAX biosensor. First, the absence of aspecific binding of each of the analysed proteins on the biosensor was verified. Next, a 90 μg.ml$^{-1}$ solution of random 30-mer DNA oligonucleotides with Biotin TEG at its 5′ extremity was

immobilised on the streptavidin SAX biosensor in $1 \times$ PBS buffer. The biosensor was further equilibrated in 20 mM HEPES pH 7.5, 150 mM potassium acetate, 2 mM magnesium acetate, 1 mM DTT, 6.3% (v/v) glycerol and 0.1% (w/v) NP-40. For the association curve, the ligand proteins were diluted at the indicated concentrations in 20 mM HEPES pH 7.5, 150 mM potassium acetate, 2 mM magnesium acetate, 1 mM DTT, 6.3% (v/v) glycerol and 0.1% (w/v) NP-40 and further incubated with the SAX biosensor. The following dissociation curve was obtained by incubating the biosensor in 20 mM HEPES pH 7.5, 150 mM potassium acetate, 2 mM magnesium acetate, 1 mM DTT, 6.3% (v/v) glycerol and 0.1% (w/v) NP-40 without ligand. The biosensor was regenerated with buffer containing 150 mM NaCl, 1.25 mM EDTA and 0.125% SDS. The $k_{on}$, $k_{off}$ and $K_D$ were calculated using the manufacturer's software (BLItz Pro 1.2). Each experiment was conducted at least in triplicate.

The affinity of UPF1-HD WT and mutant proteins for GST-Tax was analysed with an AR2G sensor (ForteBio). First, GST-Tax was covalently bound to the biosensor following the manufacturer's procedure: the biosensor was activated for 300 s with 20 mM EDC, 10 mM NHS and 10 mM sodium acetate pH 5. Next, 20 nM of Tax diluted in 100 mM sodium acetate pH 4 were linked for 500 s before quenching with 1 M ethanolamine pH 8.5. The baseline was acquired after 120 s of incubation in the manufacturer's running buffer (300 mM NaCl, 20 mM phosphate, 0.02% Tween 20, 0.1% albumin, 0.05% ProClin300) before 250 s of incubation with UPF1 His-HD protein diluted in the running buffer. Finally, the dissociation step was acquired after 60 s of incubation of the sensor with running buffer without protein.

**Experiment with magnetic tweezers**. The DNA hairpin used for single-molecule experiments was performed as described by Fiorini et al.[33] The 1.2-kbp DNA substrate is a 1239-bp hairpin with a 4-nt loop, a 76-nt 5′-biotinylated ssDNA tail and a 146 bp 3′-digoxigenin-labelled dsDNA (sequence available in supplementary methods). We used a PicoTwist magnetic tweezers instrument (www.picotwist.com) to manipulate individual DNA. The DNA hairpins were attached by the 5′-biotinylated extremity to streptavidin-coated magnetic beads (Dynabeads MyOne streptavidin T1, Life Technology) and by a 3′-digoxigenin-modified extremity to an anti-Dig-coated glass surface. The glass coverslip had been previously treated with anti-digoxigenin antibody (Roche) and passivated with $1 \times$ PBS Buffer ($1 \times$ PBS pH 7.5, 0.2% pluronic surfactant, 5 mM EDTA, 10 mM sodium azide and 0.2% BSA (Sigma-Aldrich)). Experiments were conducted at 37 °C in helicase buffer (20 mM Tris-HCl pH 7.5, 75 mM potassium acetate, 3 mM magnesium chloride, 2% BSA, 0.5 mM DTT and 2 mM ATP). The indicated UPF1 concentration was the lowest possible concentration to observe helicase activity under single-molecule conditions (between 1 and 20 nM depending on the protein batch). $Tax_{CACA}$ was used at a 5 nM final concentration.

**Data availability**. All relevant data are available from the authors.

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

## Acknowledgements

We would like to thank Saurabh Raj at LPS of ENS-Paris for assistance during MT data collections and Derse's lab for pCMV-HTLV-1 plasmids. We also thank Armelle Roisin, Sébastian Durand and Stéphane Rety at LBMC of ENS-Lyon for technical support and for suggestions and Christophe Guillon at MMSB-Lyon and Vincent Vanoosthuyse at LBMC-Lyon for useful comments on the manuscript. We acknowledge the contribution of SFR Biosciences (UMS3444/CNRS, US8/Inserm, ENS de Lyon, UCBL) facilities: PLATIM and PSF, in particular Véronique Senty-Ségault for help in BLItz experiments' set-up. Work in the authors' laboratory is supported by grants from foundation ARC (V. M.), from La Ligue Isere contre le Cancer (V.M.) and French Agence Nationale de Recherche sur le SIDA et les Hépatite Virales (ANRS) fellowship (F.F.).

## Author contributions

Protein purifications, pulldown and enzymatic bulk assays: F.F.; protein purifications and BLItz experiments: J.-P.R.; Helicase MT assays: J.K., F.F., H.L.H. and V.C.; immuno-fluorescence assays: M.B.; pull-down assays and ex vivo experiments: V.M.; writing the manuscript: F.F., V.M. and P.J.

## Additional information

**Competing interests:** The authors declare no competing financial interests.

