## [Peer Review File · Nature Communications]

Reviewers' comments:

Reviewer #1 (Remarks to the Author):

Fiorini et al. further study the mechanism of inhibition of NMD by the HTLV-I Tax protein. In a previous paper from this group (Mocquet et al. JVI 2012), they showed that Tax does this by interacting with both UPF1 and INT6/EIF3E. Here, they focus on the interaction of Tax with UPF1. Interestingly, they find that Tax reduces the ability of UPF1 to bind nucleic acids and to hydrolyze ATP. Further, Tax blocks translocation of UPF1, suggesting the importance of this process for NMD. This is novel and very interesting work, using a sophisticated single molecule approach.

Comments:

1. This paper needs editing by a native English speaker. There are numerous awkward phrases, including the following from the Introduction. There are additional problems throughout.

line 79: "role in pervasive and cryptic transcripts decay in yeast, appearing as major player" should be rewritten as "role in the pervasive and cryptic decay of transcripts in yeast, appearing as a major player"

line 83: "UPF1 undergoes SMG1-mediated multiple phosphorylations" should be "UPF1 undergoes multiple SMG1-mediated phosphorylations, "

line 96: rising should be raising

line 98: "3' mRNP remodeling and decay completion" should be "completion of 3'MRNP remodeling and decay"

line 106: "There are now many evidences" should be There is now much evidence

2. Do the authors still think Tax needs to bind to INT6/EIF3E in addition to UPF1 to inhibit NMD?

3. Line 137-138: what is the reference for this?

Reviewer #2 (Remarks to the Author):

Referee report for "Retroviral Tax plugs and freezes UPF1 helicase leading to NMD inhibition during HTLV-1 infection" by Fiorini et al.

In their manuscript Fiorini et al. study the interaction/inhibition of the UPF1 helicase with the retroviral protein Tax, using a number of bulk biochemical assays and single-molecule magnetic tweezers. The helicase UPF1 is a key player in nonsense-mediated mRNA decay

and has been investigated in a number of previous studies, including work at the single-molecule level by the same authors.

The author demonstrate convincingly that Tax interacts with UPF1 and that it inhibits/reduces its affinity to RNA/DNA (also a change of ~10-fold is not exactly dramatic?). Using magnetic tweezers, the authors show that this is even the case for UPF1 actively translocating on DNA.

Overall, these results lend support to a mechanism by which the retrovirus protects its genome from degradation and possibly also interferes in cellular pathways more generally.

The work is well executed, but it is sometimes a little hard to follow what the novel findings are and what their relevance is. This should be more clearly articulated. In addition, I have a number of more specific comments:

- Language. The language is sometimes confusing and should be improved. Specific examples are included below.

For one specific and instructive example, consider the title: It describes the action of Tax as "plugs and freezes", neither of which are technical terms normally used in this context and neither of which occurs anywhere else in the manuscript. In addition, the title contains three not very commonly used acronyms ("UPF1", "NMD", and "HTLV-1").

- Are the differences in decay rates shown in Figure 1 significant? What statistical tests were done to ascertain this?

Minor points / language:

- Abstract: "functioning" -> why not just "function"?

- Abstract: "[...] retroviral infection of Human T-cell lymphotropic virus type I (HTLV-1)" Do the authors mean "infection of" or "infection by"? I suspect the later, even though the former is written.

- Abstract: "by using single molecule approach" -> Something is missing.

- Introduction, page 4: What is meant by "release" in "Stabilized at the stop codon, UPF1 then successfully releases its essential ATPase activity"?

- When the "modular enzyme" is discussed (Intro, page 5, line 91), refer to schematic?

- Intro, page 5, line 106: "There are now many evidences" sounds strange. Rephrase.

- Intro, page 6, line 117: "at translation terminating ribosome to prevent" -> Something is missing?

- Intro, page 6, line 125: "prompting the importance of this mechanical feature for NMD process." Sounds strange. In particular "prompting" seems out of place?
- Page 6, line 135 "harboring or not" sounds strange. Rephrase?
- Page 8, line 181: "Several point mutations, whose position is" why not plural "positions are"...?
- Page 11, line 252: "lower affinity" -> Compared to what? Complete comparatives!
- Page 11, line 260: "We deemed" seems to be used in a strange way here?
- Page 15, line 366: "UPF1 ATPase activity owns" seems to have something missing and/or a strange used of "owns".
- Figure 1: Many of the labels here are very small and hard to read.
- Figure 3c: Add multiplication dots, where appropriate.

Reviewer #3 (Remarks to the Author):

Evaluation of the manuscript "Retroviral Tax plugs and freezes UPF1 helicase leading to NMD inhibition during HTLV-1 infection" by Vincent Mocquet and co-workers. Nonsense-mediated mRNA decay is an important cellular pathway that detect and eliminates faulty mRNAs and prevents the expression of C-terminally truncated proteins. Several transacting factors are required for efficient NMD, amongst them the central NMD protein UPF1. UPF1 is a conserved RNA helicase, which also exhibits RNA-dependent ATPase activity. The helicase activity of UPF1 is required for NMD, but the precise molecular processes occurring during the detection and the degradation of NMD substrate mRNAs are only partially understood.

In this manuscript, Fiorini et al. analyze the inhibition of NMD by the viral protein Tax (derived from human T-cell lymphotropic virus type I). They used a combination of different assays to show that Tax interacts with the helicase domain of UPF1. The main conclusion of the manuscript is that Tax: (a) binds to UPF1 and prevents the binding to substrate transcripts and (b) negatively influences the unwinding and/or ATPase activity of UPF1. The manuscript is a follow-up work of a previous paper, in which the authors reported the interaction of Tax with the cellular NMD machinery and the inhibition of NMD (Mocquet et al., 2012, PMID: 22553336). This manuscript provides some new information regarding the NMD-inhibitory function of Tax, for which the underlying experiments were performed in an acceptable quality. However, many conclusions with respect to NMD go too far and are not properly controlled. Furthermore, the manuscript contains several flaws and inconsistencies, which are listed below.

General comments:

1. Despite the apparent effect of Tax on PTC-containing mRNA, it is not sufficiently

demonstrated that the observed reduction of UPF1-RNA binding or decrease of UPF1 helicase activity are the cause for the NMD inhibition. A more detailed discussion of this point can be found below.

2. Many conclusions are not properly substantiated or the connection of the conclusion with the experimental results is questionable. For example, Tax inhibits the ATPase activity of UPF1 in vitro (Fig 2d). How does this connect to the reduced mRNA binding, as stated in the discussion (line 380-386)? Other examples are examined in more detail below.

Major specific comments:

3. Figure 1: Considering that the earlier publication of the lab already described the NMD-inhibiting effect of Tax, the Figure 1 does not provide additional insight in the mechanism of Tax-mediated NMD inhibition.

4. Supplementary Figure 1a: The northern blot analysis raises several questions:

a. Why are the overall steady-state levels of PTC-containing globin mRNA reporter lower when Tax is overexpressed (compare lanes 1 and 5)? If indeed NMD is inhibited, which is inferred by the increased half-life of the PTC reporter, then the reporter should accumulate. By comparing the expression of GI-WT GAPDH and GI-PTC directly, it seems unlikely that expression of plasmid-derived reporter mRNA in general is disturbed when Tax is expressed. The authors should therefore explain why the initial PTC-reporter levels are reduced upon Tax expression. Furthermore, the expression levels of reporter mRNAs have to be shown for all NMD assays.

b. As shown in Supplementary Figure 1c and the previous publication by the same authors, the presence of Tax increases P-body foci. Furthermore, UPF1 seems to accumulate in these foci (Mocquet et al., 2012, PMID: 22553336) upon Tax expression. Additionally, it was shown that mRNAs mainly accumulate in P-bodies upon leaving the translation cycle (PMID: 17403906). A simple explanation for the apparent increase in PTC-RNA half-life could be the "storage" of these RNAs in P-bodies. Since NMD is a translation-dependent process, this could explain why the PTC-mRNAs are no longer rapidly degraded. Have the authors considered this hypothesis? Along the same line, what is the impact of Tax expression on overall translation or translation of PTC-containing transcripts?

5. Figure 1 and line 546-547: Why were different amounts of globin reporter (WT/PTC) transfected for the qRT-PCR experiments?

6. Figure 1: Which primer pair (globin 3' or FL) was used for the qRT-PCR in Figure 1? What is the purpose in general for those two primer pairs?

7. Figure 1: What is the impact of Tax expression on other NMD target classes such as alternatively spliced NMD targets or mRNAs with "long" 3' UTR?

8. Figure 1: The half-life measurements are shown in Ln (N/No) vs time plots, which are inherently difficult to extract information from. Furthermore, how is the time constant λ defined and calculated for the half-life measurements?

9. Figure 1 and line 561: What is meant by "... three independent measures ..."? Are these biological replicates or technical replicates?

10. Figure 2d: An important point of the manuscript is the modulation of UPF1 ATPase activity upon Tax binding, which is shown in Figure 2d. Given the decrease of UPF1 binding to RNA upon interacting with Tax (Figure 3-4), it is unclear whether the impaired ATP hydrolysis is the cause or rather the consequence, since UPF1 is an RNA-dependent ATPase. This problem needs to be addressed.

11. Figure 2c and lines 201-209: Considering Figure 2b, in which the interaction of GST-Tax

with UPF1 proteins was already shown, what exactly is the purpose of the paragraph L. 201-209 and the corresponding Figure 2c? If indeed the presence of UPF1 helps to stabilize GST-Tax, why were both proteins not expressed simultaneously in bacteria and co-purified, instead of performing co-lysis of separately cultured bacteria? A clear and consistent experimental setup should be used throughout the manuscript.

12. Figure 2: What is the impact of the C23/C29 mutation on the ability to inhibit NMD? Does the ZnF domain or the unspecific RNA binding influence this? A functional test of this mutant has to be shown.

13. Figure 3: Does Tax expression or HTLV infection in general lead to decreased UPF1 binding to overall mRNA, not just the PTC-containing globin? Since UPF1 was shown to bind non-specifically to mRNA, Tax could prevent this "general" binding as well. This should be tested.

14. Figure 3: The usage and interpretation of the dominant-negative UPF1 mutant R843C is problematic. The authors should use other mutants of UPF1 (and preferentially also Tax) to impair the interaction of UPF1 and Tax.

15. Supplementary Figure 3e: Binding of Tax to the UPF1-HD mutant (R843C) seems to be only slightly reduced. Therefore it will be mandatory to determine binding affinities of Tax to WT and mutant UPF1.

Minor points:

16. Line 80: The wrong reference (6) is used here and needs to be corrected.

17. Line 137-138: The authors write that "Due to a 5' uORF, GADD45a is one of the most destabilized mRNA by the NMD process in both Drosophila and mammalian cells", but do not include a reference for this statement. To which other NMD targets was that comparison made?

18. Supplementary Figure 3e: It is not acceptable to remove input samples before the addition of Tax, because the ratio of UPF1 and Tax cannot be estimated.

19. Knockdown by siRNA transfection (siCtr and siUPF1) was performed but not described in the Methods section.

20. General comments on spelling and style:

Lines 138-139: In order to determine their stability, a monitoring of GI-PTC, GI-WT and GADD45a mRNAs level was performed using qRT-PCR..." – please revise

Lines 369-372: Complicated sentence, please revise

Line 389: slightly decreased

Line 468: PCR products

Line 480: kindly Line 489: bacteria

Line 489 and 697: overnight

Line 515: Coomassie

Line 551: transcription

Line 677 and 703: set

The space between number and unit was not consistently used (e.g. line 489: "... 1L of LB." and line 491: "... on 1 ml of ...")

Answer to Reviewers' comments:

Reviewer's 1 comments 1 and 3, Reviewer's 2 comment 1 and Reviewer's 3 comments 16 and 20:

We are very grateful to the Reviewers for their detailed comments concerning reference and language inaccuracies or errors of the first version of the manuscript.

All these errors have been corrected and the English language usage of the revised version has been edited by a professional service.

Reviewer #1 (Remarks to the Author):

Fiorini et al. further study the mechanism of inhibition of NMD by the HTLV-I Tax protein. In a previous paper from this group (Mocquet et al. JVI 2012), they showed that Tax does this by interacting with both UPF1 and INT6/EIF3E. Here, they focus on the interaction of Tax with UPF1. Interestingly, they find that Tax reduces the ability of UPF1 to bind nucleic acids and to hydrolyze ATP. Further, Tax blocks translocation of UPF1, suggesting the importance of this process for NMD. This is novel and very interesting work, using a sophisticated single molecule approach.

Comments:

2. Do the authors still think Tax needs to bind to INT6/EIF3E in addition to UPF1 to inhibit NMD?

The role of INT6 during NMD is still incompletely understood. It was suggested that UPF2 and CBP 80 might interact with INT6 through their MIF4G domains (Morris et al. 2007 PMID: 17468741). Before addressing further how Tax interaction with INT6 intervenes in NMD inhibition, we are currently working to better understand the interplay between this eIF3 subunit and the other key factors participating to NMD. An important issue is in particular to determine at which step INT6 is important in the process.

Reviewer #2 (Remarks to the Author):

Referee report for "Retroviral Tax plugs and freezes UPF1 helicase leading to NMD inhibition during HTLV-1 infection" by Fiorini et al.

In their manuscript Fiorini et al. study the interaction/inhibition of the UPF1 helicase with the retroviral protein Tax, using a number of bulk biochemical assays and single-molecule magnetic tweezers. The helicase UPF1 is a key player in nonsense-mediated mRNA decay and has been investigated in a number of previous studies, including work at the single-molecule level by the same authors.

The author demonstrate convincingly that Tax interacts with UPF1 and that it inhibits/reduces its affinity to RNA/DNA (also a change of ~10-fold is not exactly dramatic?). Using magnetic tweezers, the authors show that this is even the case for UPF1 actively translocating on DNA. Overall, these results lend support to a mechanism by which the retrovirus protects its genome from degradation and possibly also interferes in cellular pathways more generally.

The work is well executed, but it is sometimes a little hard to follow what the novel findings are and what their relevance is. This should be more clearly articulated. In addition, I have a number of more specific comments:

2. For one specific and instructive example, consider the title: It describes the action of Tax as “plugs and freezes”, neither of which are technical terms normally used in this context and neither of which occurs anywhere else in the manuscript. In addition, the title contains three not very commonly used acronyms (“UPF1”, “NMD”, and “HTLV-1”).

We acknowledge the Reviewer’s argument but we also feel that the title should truly convey the complexity of the Tax-mediated UPF1 control. Indeed Tax has a double effect on UPF1 activity: Tax binds the UPF1 region located at the entry site of RNA, hindering its binding to the substrate (plugging effect) and Tax can also associate with actively running UPF1, thereby blocking its translocation activity (freezing effect). Stating this information explicitly (in title and abstract) is also likely to attract more attention to the paper. We would like to have your editorial opinion on this matter. We would rather keep the title as it is but, if you concur with the Reviewer, we may change it to something more restrictive like **“Mechanistic insight into Tax-mediated dual inhibition of UPF1 RNA binding and translocation during HTLV-1 infection”**.

3. Are the differences in decay rates shown in Figure 1 significant? What statistical tests were done to ascertain this?

The figure 1 shows the decay curves representing the average of at least 3 independent biological replicates. The half-lives were calculated for each replicate and a Student’s t-test (unpaired, two tailed) was conducted.

Notably, we reorganized the figure 1 in order to highlight the new data compared to the previous manuscript (Mocquet et al. 2011 PMID: 22553336). This organization also allows us to answer Reviewer’s 3, point 3.

We eliminated the figure 1a and displaced 1c to supplementary figure 1a. We kept figure 1b (actual 1a), 1d (actual 1b) e and f (actual 1c and d respectively). Notably, we performed two additional experiments in order to test endogenous NMD-prone mRNAs in addition to GADD45 α . The decay assays of SMG5 and MAP3K14 mRNAs are now shown in figure 1e and 1f, respectively. With different NMD-inducing features these observations confirm what was observed with GADD45 α .

Moreover, in order to increase the clarity of the statistical data, the p values of t-tests for each experimental condition were indicated below graphs within the legends. In these tests, we considered two levels of significance, indicated by * or **, when the p value is lower than 0.05 or 0.01, respectively.

Reviewer #3 (Remarks to the Author):

Evaluation of the manuscript “Retroviral Tax plugs and freezes UPF1 helicase leading to NMD inhibition during HTLV-1 infection” by Vincent Mocquet and co-workers. Nonsense-mediated mRNA decay is an important cellular pathway that detect and eliminates faulty mRNAs and prevents the expression of C-terminally truncated proteins. Several transacting factors are required for efficient NMD, amongst them the central NMD protein UPF1. UPF1 is a conserved RNA helicase, which also exhibits RNA-dependent ATPase activity. The helicase

activity of UPF1 is required for NMD, but the precise molecular processes occurring during the detection and the degradation of NMD substrate mRNAs are only partially understood. In this manuscript, Fiorini et al. analyze the inhibition of NMD by the viral protein Tax (derived from human T-cell lymphotropic virus type I). They used a combination of different assays to show that Tax interacts with the helicase domain of UPF1. The main conclusion of the manuscript is that Tax: (a) binds to UPF1 and prevents the binding to substrate transcripts and (b) negatively influences the unwinding and/or ATPase activity of UPF1.

The manuscript is a follow-up work of a previous paper, in which the authors reported the interaction of Tax with the cellular NMD machinery and the inhibition of NMD (Mocquet et al., 2012, PMID: 22553336). This manuscript provides some new information regarding the NMD-inhibitory function of Tax, for which the underlying experiments were performed in an acceptable quality. However, many conclusions with respect to NMD go too far and are not properly controlled. Furthermore, the manuscript contains several flaws and inconsistencies, which are listed below.

General comments:

1. Despite the apparent effect of Tax on PTC-containing mRNA, it is not sufficiently demonstrated that the observed reduction of UPF1-RNA binding or decrease of UPF1 helicase activity are the cause for the NMD inhibition. A more detailed discussion of this point can be found below.

We thank Reviewer 3 for this fair comment. The discussion on this point has been broadened in the manuscript. Actually, the role of RNA-affinity and enzymatic activity of UPF1 for the correct completion of NMD is extensively described in the literature (Weng et al. 1996 PMID: 8816461). Lykke-Andersen and co-workers showed that disassembly and completion of turnover of mRNPs undergoing NMD requires ATP hydrolysis by UPF1 (Franks et al 2010 PMID: 21145460). Moreover, Baker and co-workers demonstrated that the complete degradation of PTC-harboring mRNAs required proper ATPase activity and RNA binding properties of UPF1 (Serdar et al. 2016 PMID: 28008922). The ATPase activity is also critical for NMD target discrimination (Lee et al. 2015 PMID : 26253027). As a consequence, we consider that all factors or mutations able to prevent the enzymatic activity of UPF1 also affect the NMD process.

2. Many conclusions are not properly substantiated or the connection of the conclusion with the experimental results is questionable. For example, Tax inhibits the ATPase activity of UPF1 *in vitro* (Fig 2d). How does this connect to the reduced mRNA binding, as stated in the discussion (line 380-386)?

The Reviewer is right, the sentence might be misinterpreted. UPF1 is a nucleic acid-dependent ATPase and we demonstrate that the effect of Tax on UPF1 ATPase activity was due to an RNA binding defect rather than a missing ATP binding (described in results section). The sentence in line 380-386 has been changed and now reads: "We show that Tax strongly affects the ATPase activity of UPF1 leaving its ATP binding ability unchanged (Fig. 2d and e). While UPF1 is a nucleic acid-dependent hydrolase, we characterized two different ways of Tax-mediated control of its activity, depending on the moment Tax interacts with UPF1.

Other examples are examined in more detail below.

Major specific comments:

3. Figure 1: Considering that the earlier publication of the lab already described the NMD-inhibiting effect of Tax, the Figure 1 does not provide additional insight in the mechanism of Tax-mediated NMD inhibition.

As answered for Reviewer's 2, point 3, we changed the figure 1 as described above. In this manuscript, we showed the role of Tax in the context of a complete HTLV-1 molecular clone on both reporter and endogenous NMD-sensitive mRNAs. We believe that by using HTLV-1 molecular clone, we are closer to the physiological condition as compared with the use of a vector expressing solely Tax. This latter experiment has been moved to supplementary figure 1a as control.

4. Supplementary Figure 1a: The northern blot analysis raises several questions:

a. Why are the overall steady-state levels of PTC-containing globin mRNA reporter lower when Tax is overexpressed (compare lanes 1 and 5)? If indeed NMD is inhibited, which is inferred by the increased half-life of the PTC reporter, then the reporter should accumulate. By comparing the expression of GI-WT GAPDH and GI-PTC directly, it seems unlikely that expression of plasmid-derived reporter mRNA in general is disturbed when Tax is expressed. The authors should therefore explain why the initial PTC-reporter levels are reduced upon Tax expression. Furthermore, the expression levels of reporter mRNAs have to be shown for all NMD assays.

The Reviewer is right, it is true that GI-PTC steady state levels are lower when Tax is expressed. To explain this point, we have to consider that Tax is a powerful transcription modulator: it can enhance as well as inhibit gene expression depending on the promoter and on the molecular partners involved. It is then likely that the steady state levels of GI-PTC were transcriptionally as well as post-transcriptionally modulated by Tax. For this reason we favor the analysis of mRNA half-lives rather than mRNA steady state levels to measure NMD inhibition. Moreover, the different level of GI-PTC and GI-WT mRNAs can also be explained considering their different promoters (tetracyclin inducible promoter vs CMV respectively).

The northern blot in the supplementary figure 1b has the vocation to validate the qRT-PCR performed for decay assays and confirm the trend of the results showed.

b. As shown in Supplementary Figure 1c and the previous publication by the same authors, the presence of Tax increases P-body foci. Furthermore, UPF1 seems to accumulate in these foci (Mocquet et al., 2012, PMID: 22553336) upon Tax expression. Additionally, it was shown that mRNAs mainly accumulate in P-bodies upon leaving the translation cycle (PMID: 17403906). A simple explanation for the apparent increase in PTC-RNA half-life could be the "storage" of these RNAs in P-bodies. Since NMD is a translation-dependent process, this could explain why the PTC-mRNAs are no longer rapidly degraded. Have the authors considered this hypothesis? Along the same line, what is the impact of Tax expression on overall translation or translation of PTC-containing transcripts?

We acknowledge the Reviewer for providing this consistent hypothesis. In the discussion section, we suggest that the translocation inhibition of UPF1 induced by Tax, might block the late steps of the NMD process, preventing 3'RNA end fragment decay and inducing their accumulation in P-

bodies with their associated protein complexes including UPF1 and Tax (Mocquet et al. 2012 PMID: 22553336). To address this Reviewer's point, we tested the effect of Tax expression on bulk translation. We carried out a metabolic labelling using 35S-methionine and we measured incorporation of radioactivity within the cellular proteins as described in Legros et al., 2011 (PMID: 21532619). This new experiment shows that Tax does not significantly modify the overall level of newly-synthesized cellular proteins. This result is shown in supplementary figure 1f of this revised version. In their previous report Legros and coworkers similarly observed in HeLa cells without arsenic stress, 35S-methionine incorporation was not significantly modified under Tax expression.

Thus it appears that the Tax impact on RNA stability does not correlate with a general translation inhibition.

5. Figure 1 and line 546-547: Why were different amounts of globin reporter (WT/PTC) transfected for the qRT-PCR experiments?

Under normal conditions, the steady state levels of GI-PTC RNA are around 10-20 times lower than those of GI-WT RNA. Thus we decided to transfect less GI-WT reporter in order to equilibrate the levels of globin RNAs at the starting point of the decay or RIP analysis to be able to compare the results.

6. Figure 1: Which primer pair (globin 3' or FL) was used for the qRT-PCR in Figure 1? What is the purpose in general for those two primer pairs?

We only used the primer pair globin3' (both oligos on the 3' side of the PTC) for the RNA decay assays monitoring the globin RNA by qRT-PCR in figure 1. The purpose of the primer pair "FL" (oligos encompassing the PTC position) was to validate the UPF1 RIP experiments (Fig. 3a) with another set of primers. The presentation of the "FL" primers in figure 1 was effectively confusing; we removed it. All the primer sequences are provided in the methods section.

7. Figure 1: What is the impact of Tax expression on other NMD target classes such as alternatively spliced NMD targets or mRNAs with "long" 3' UTR?

We performed new decay assays using two additional endogenous NMD substrates (results are shown in figure 1e and 1f). We used SMG5 mRNA because it presents a long 3'-UTR and it is a well-known NMD substrate in mammalian and *Drosophila* (Yepiskoposyan et al. 2011 PMC3222124; Rehwinkel et al. 2005; Mendell et al. 2004; Chan et al. 2007). We also used MAP3K14 mRNA (Mendell et al. 2004). Based on the results of the German sequencing consortium, there is an alternative 5' exon donor in exon11, leading to an out of frame translation and a premature termination codon more than 55 nt upstream an EJC, making this RNA NMD sensitive (genbank: CR749592.1 from clone DKFZp686J04131 cDNA sequencing consortium of the German Genome Project).

8. Figure 1: The half-life measurements are shown in Ln (N/No) vs time plots, which are inherently difficult to extract information from. Furthermore, how is the time constant λ defined and calculated for the half-life measurements?

An exponential decay is described by the following formula :

$N(t) = N_0 e^{-\lambda t}$ i.e. $\ln(N/N_0) = -\lambda \cdot t$. (J. G. Belasco and G. Brawerman, Control of Messenger RNA Stability, 2012) where N_0 is the initial amount of mRNA, $N(t)$ is the amount of mRNA that remains at time t , and λ the decay constant. As indicated in the legend section, the half-life ($t_{1/2}$) is then given by $t_{1/2} = \ln(2)/\lambda$.

By representing half-lives measurement in $\ln(N/N_0)$ vs time, the decay curves became a linear function following the equation: $\ln(N/N_0) = -\lambda \cdot t$ what is much easier to extract λ . Those linear functions are the one presented in the graphs of figure 1.

9. Figure 1 and line 561: What is meant by "... three independent measures ..."? Are these biological replicates or technical replicates?

The decay curves in figure 1 represent the average of at least 3 independent biological replicates.

10. Figure 2d: An important point of the manuscript is the modulation of UPF1 ATPase activity upon Tax binding, which is shown in Figure 2d. Given the decrease of UPF1 binding to RNA upon interacting with Tax (Figure 3-4), it is unclear whether the impaired ATP hydrolysis is the cause or rather the consequence, since UPF1 is an RNA-dependent ATPase. This problem needs to be addressed.

The reviewer is right. We demonstrated that Tax interaction to UPF1 does not affect ATP binding but lowers RNA affinity and decreases the ATP hydrolysis activity. We conclude that the low RNA binding is most likely the cause of the loss of enzymatic activity. This point has been stressed in results and discussion paragraphs.

11. Figure 2c and lines 201-209: Considering Figure 2b, in which the interaction of GST-Tax with UPF1 proteins was already shown, what exactly is the purpose of the paragraph L. 201-209 and the corresponding Figure 2c? If indeed the presence of UPF1 helps to stabilize GST-Tax, why were both proteins not expressed simultaneously in bacteria and co-purified, instead of performing co-lysis of separately cultured bacteria? A clear and consistent experimental setup should be used throughout the manuscript.

The reviewer is right, the simultaneous expression of two interacting proteins is the best way to obtain an operative complex. Moreover, Tax is predicted to have extended unstructured regions and we made the hypothesis that UPF1 may help to stabilize the right conformation avoiding toxicity for bacteria, precipitation and/or protein degradation. Indeed, we co-expressed both UPF1-HD and Tax proteins from the same polycistronic construct but this method yielded several contaminants and lower amounts of complex compared to the co-lysis technique. In the experiment showed in the figure, Upf1-HD was fused to a C-terminal Histidine tag (HIS) while Tax-C (aa 178-350) contained a N-terminal Calmodulin-binding peptide tag (CBP). Sequential purifications on Nickel and calmodulin resins yielded a bimolecular complex demonstrating the interaction of UPF1-HD with this part of Tax but the amount and the purity of the preparation was inadequate for functional tests. We decided to go for a co-lysis experiment (Fig. 2c), which was successfully used elsewhere (Chakrabarti et al. 2011 PMID: 21419344; Fiorini et al. 2013 PMID: 23275559), to obtain the suitable UPF1-Tax complex.

As a consequence, the meaning of figure 2c is to present how the HD-Tax complex used throughout this whole story has been purified.

As a consequence, the meaning of figure 2c is to present how the HD-Tax complex used throughout this whole story has been purified.

12. Figure 2: What is the impact of the C23/C29 mutation on the ability to inhibit NMD? Does the ZnF domain or the unspecific RNA binding influence this? A functional test of this mutant has to be shown.

To answer this point we constructed a derivative of the pSG5 vector to express the mutant Tax_{CACA} in mammalian cells and tested its effect on NMD using the GI-PTC reporter. By performing a decay assay we observed that as compared to control conditions (pSG5, t(1/2)= 2.2h) both Tax_{CACA} mutant and Tax wt stabilized the PTC-containing mRNA, leading to similar half-lives (t(1/2)= 4.2h for Tax_{CACA} and t(1/2)= 4.7h for Tax wt – Supplementary Fig. 1a).

Thus, the mutation C23AC29A does not modify Tax ability to inhibit NMD *ex vivo*.

This result was added in our manuscript as supplementary figure2c.

13. Figure 3: Does Tax expression or HTLV infection in general lead to decreased UPF1 binding to overall mRNA, not just the PTC-containing globin? Since UPF1 was shown to bind non-specifically to mRNA, Tax could prevent this “general” binding as well. This should be tested.

As suggested by this Reviewer we assessed the impact of Tax on UPF1 binding to the GI-WT reporter RNA in transfected HeLa cells following the same protocol as in figure. 3a. These new data are shown in supplementary figure 3a. Consistently with the biochemical data, the effect of

Tax on UPF1 RNA-binding properties can be observed with both PTC- and non-PTC-containing substrates.

14. Figure 3: The usage and interpretation of the dominant-negative UPF1 mutant R843C is problematic. The authors should use other mutants of UPF1 (and preferentially also Tax) to impair the interaction of UPF1 and Tax.

We are not sure to clearly understand this argument. The well-studied R843C mutation in the context of full length UPF1 has been shown to strongly impair NMD in yeast and mammals (Leeds et al. 1992 PMID: 1569946 ; Sun et al. 1998 PMID: 9707591; Kurosaki et al. 2014 PMID: 25184677). However, it is true that C843 might make a disulphide bridge with nearby cysteines, affecting the global folding of the helicase domain. Nevertheless, under reducing conditions, our recombinant helicase domain R843C mutant is still able to bind ATP (data not showed) and RNA even if in a lesser extent (Fig. 3e compare lines 1 and 5) indicating that the folding of the protein was not affected.

Despite this evidence, we produced a new UPF1-HD R843S mutant with the aim to avoid possible disulphide bridges and clarify the Reviewer point. We used UPF1-HD R843S mutant in new Blitz experiments showed in table of figure 3f and supplementary figure 3 i. This mutation also reduced the affinity of Tax for UPF1, suggesting that Tax binds directly arginine 843. However we believe that the Tax binding site is multipartite (see CH-domain interaction in Fig. 2b) and an accurate mapping of interactions sites on both UPF1 and Tax is needed. Towards this goal, we have undertaken the analysis of the structure of the UPF1-Tax complex.

15. Supplementary Figure 3e: Binding of Tax to the UPF1-HD mutant (R843C) seems to be only slightly reduced. Therefore it will be mandatory to determine binding affinities of Tax to WT and mutant UPF1.

In order to answer this Reviewer's question we performed a whole series of Blitz assays using UPF1-HD WT and mutants (DE633AA, R843C and R843S) to measure the UPF1-Tax affinity (new Supplementary Fig. 3 from f to i) .

We integrated the table containing the KD, Kon and Koff in figure 3f.

Minor points:

16. Line 80: The wrong reference (6) is used here and needs to be corrected.

The correct reference (Celik et al 2017 PMID: 28209632) has been inserted.

17. Line 137-138: The authors write that "Due to a 5' uORF, GADD45 α is one of the most destabilized mRNA by the NMD process in both Drosophila and mammalian cells", but do not include a reference for this statement. To which other NMD targets was that comparison made?

This sentence was modified since GADD45 α is definitively not the mRNA that is the most stabilized by NMD inhibition after UPF1 knock-down. However we decided to analyze this mRNA since recent studies in both drosophila and mammals (ref 52 and 53) highlight the importance of its NMD-dependent regulation for cell homeostasis and survival.

18. Supplementary Figure 3e: It is not acceptable to remove input samples before the addition of Tax, because the ratio of UPF1 and Tax cannot be estimated.

We agree with the Reviewer argument. While we performed Blitz assays to calculate affinity constants of UPF1 wt and mutants with Tax, and while Blitz assay is far more sensitive than GST-pull down we decided to change the supplementary figure 3e with supplementary figures 3b to j.

19. Knockdown by siRNA transfection (siCtr and siUPF1) was performed but not described in the Methods section.

The material and method section has been improved according to the Reviewer's remarks.

20. General comments on spelling and style:

See introduction of this answer to Reviewers' comments.

REVIEWERS' COMMENTS:

Reviewer #2 (Remarks to the Author):

Fiorini et al. have revised their manuscript in light of the comments and I feel that this has improved the manuscript to the point of making it suitable for publication.

Reviewer #3 (Remarks to the Author):

The manuscript is considerably improved as a result of the revisions and can be published in Nature Communications.